There are amendments to this paper

# Human eye-inspired soft optoelectronic device using high-density MoS$_2$-graphene curved image sensor array

Changsoon Choi[1,2], Moon Kee Choi[1,2], Siyi Liu[3], Minsung Kim[1,2], Ok Kyu Park[1], Changkyun Im[4], Jaemin Kim[1,2], Xiaoliang Qin[5], Gil Ju Lee[6], Kyoung Won Cho[1,2], Myungbin Kim[1,2], Eehyung Joh[1,2], Jongha Lee[1,2], Donghee Son[1,2], Seung-Hae Kwon[7], Noo Li Jeon[4], Young Min Song[6], Nanshu Lu[3,8] & Dae-Hyeong Kim[1,2]

Soft bioelectronic devices provide new opportunities for next-generation implantable devices owing to their soft mechanical nature that leads to minimal tissue damages and immune responses. However, a soft form of the implantable optoelectronic device for optical sensing and retinal stimulation has not been developed yet because of the bulkiness and rigidity of conventional imaging modules and their composing materials. Here, we describe a high-density and hemispherically curved image sensor array that leverages the atomically thin MoS$_2$-graphene heterostructure and strain-releasing device designs. The hemispherically curved image sensor array exhibits infrared blindness and successfully acquires pixelated optical signals. We corroborate the validity of the proposed soft materials and ultrathin device designs through theoretical modeling and finite element analysis. Then, we propose the ultrathin hemispherically curved image sensor array as a promising imaging element in the soft retinal implant. The CurvIS array is applied as a human eye-inspired soft implantable optoelectronic device that can detect optical signals and apply programmed electrical stimulation to optic nerves with minimum mechanical side effects to the retina.

[1] Center for Nanoparticle Research, Institute for Basic Science (IBS), Seoul, 08826, Republic of Korea. [2] School of Chemical and Biological Engineering, Institute of Chemical Processes, Seoul National University, Seoul 08826, Republic of Korea. [3] Center for Mechanics of Solids, Structures, and Materials, Department of Aerospace Engineering and Engineering Mechanics, University of Texas at Austin, 210 E 24th St, Austin, TX 78712, USA. [4] School of Mechanical and Aerospace Engineering, Seoul National University, Seoul 08826, Republic of Korea. [5] Onfea Computing LLC, 204 Jackson Street, Newton, MA 02459, USA. [6] School of Electrical Engineering and Computer Science, Gwangju Institute of Science and Technology, Gwangju 61005, Republic of Korea. [7] Division of Bio-imaging, Korea Basic Science Institute, Chun-Cheon 24341, Republic of Korea. [8] Department of Electrical and Computer Engineering, Department of Biomedical Engineering, Texas Materials Institute, the University of Texas at Austin, Austin, TX 78712, USA. Changsoon Choi, Moon Kee Choi, and Siyi Liu contributed equally to this work. Correspondence and requests for materials should be addressed to N.L. (email: nanshulu@utexas.edu) or to D.-H.K. (email: dkim98@snu.ac.kr)

Soft bioelectronic devices[1–3], employing soft materials[4–6], and/or ultrathin device designs[7–9] have attracted significant attention particularly in implantable device applications[10]. For instance, the silicone-encapsulated soft neural implant effectively stimulates spinal cord to rehabilitate the disabled leg[1], and ultrathin prosthetic skin connected to peripheral nerves perceives external mechanical/thermal signals and transfers the corresponding signals to brain[11, 12]. Likewise, the soft bioelectronic device can have an important role in the intraocular retinal prosthesis for patients with retinal degeneration (e.g., macular degeneration or retinitis pigmentosa). As the optic nerves widely spread in the soft (~ 20 kPa)[13] and hemispherically shaped retina, a soft and curved form of the high-density image sensor and electrode array which mechanically matches with the human retina is significantly needed particularly for the long-term retinal prosthesis. Conventional wafer-based rigid and planar imaging modules, however, are far from this goal because lamination of planar devices can cause the retinal deformation[14], stiff devices can damage the non-regenerative optic nerves[15], and bulky multi-lens optics is required to focus on the flat image sensor (Supplementary Fig. 1a).

Recently, new image sensor arrays based on novel materials[16, 17] and device designs[18–21] have been proposed. Among these, hemispherically curved image sensor (CurvIS) arrays have gained particular attention, as they can achieve the aberration-free imaging[18] (Supplementary Fig. 1b, 2) and the wide field-of-view[19] (Supplementary Fig. 1c). These CurvIS arrays have employed distinctive interconnect designs (e.g., pop-up[18] and/or serpentine-shaped[19] structures) to absorb bending induced strains in the rigid silicon-based photodetector array. However, these interconnect designs take space and hence limit the density of the image sensor array (Supplementary Fig. 3). A relatively high-density cylindrical silicon image sensor array was reported[20], but the unidirectionally curved imager cannot provide all benefits of the omnidirectionally curved system[18, 19]. Ultrathin MoS2[22], an inherently soft two-dimensional (2D) nanomaterial[23, 24], is a promising candidate of a photo-absorbing component in the high-density omnidirectional CurvIS array attributing to its

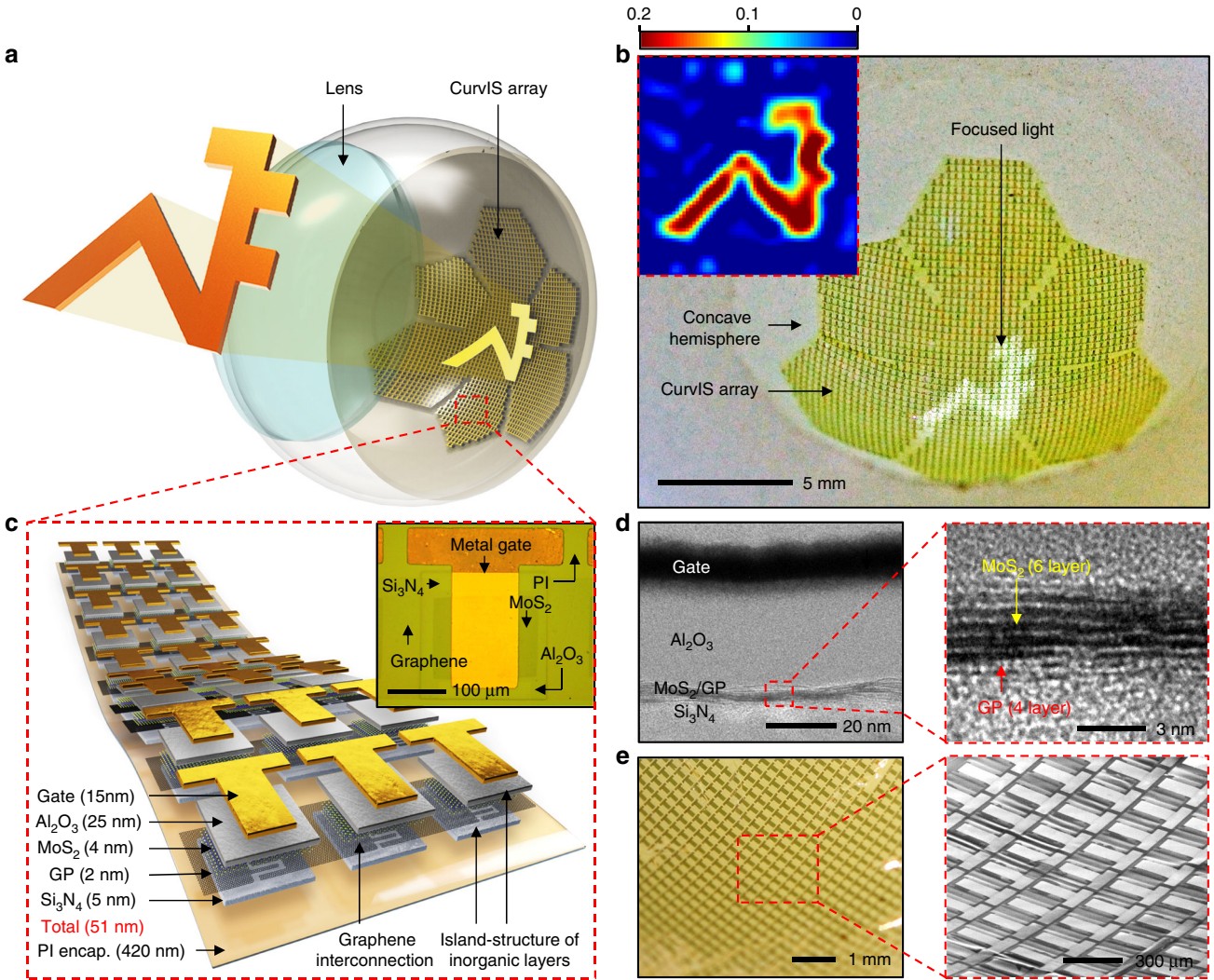

**Fig. 1** High-density curved image sensor array based on the $MoS_2$-graphene heterostructure. **a** Schematic illustration of the high-density CurvIS array based on the $MoS_2$-graphene heterostructure. **b** Optical camera image of the high-density CurvIS array. Inset shows the image (i.e., university logo) captured by the CurvIS array. **c** Schematic illustration of the device design. Inset shows an optical microscope image of a single phototransistor. **d** Cross-sectional transmission electron microscope image of the $MoS_2$-graphene phototransistor (left) and the magnified image of the $MoS_2$-graphene heterostructure (right). **e** Optical (left) and magnified scanning electron microscope (right) image of the high-density CurvIS array on the concave hemisphere

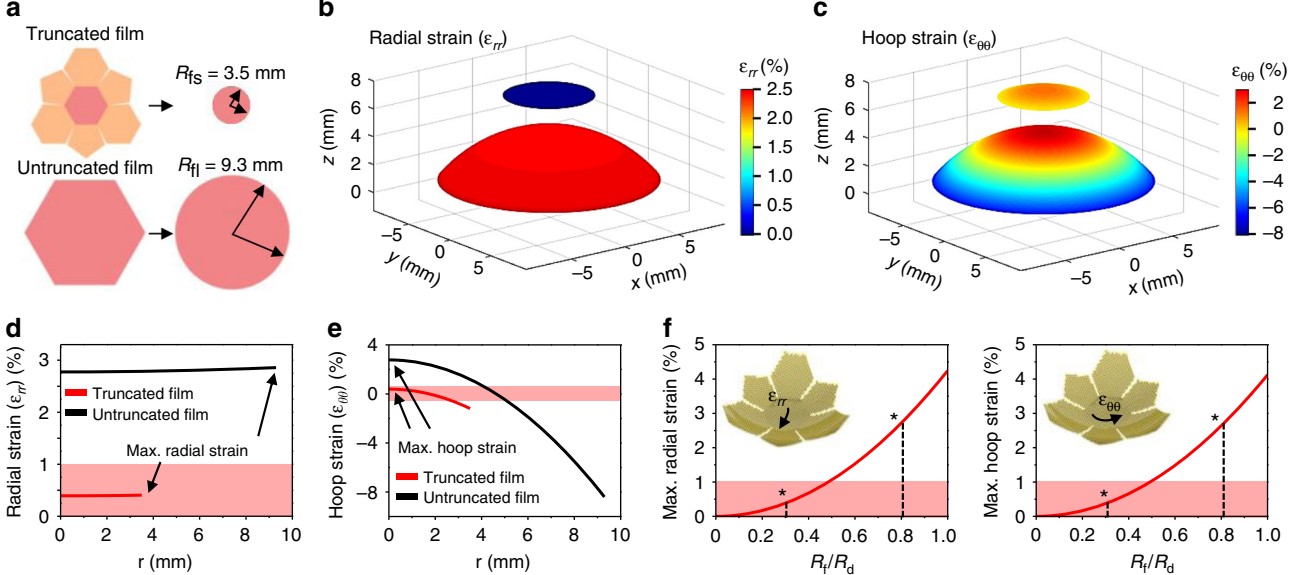

**Fig. 2** Theoretical analysis of the induced strain on the device array conformed to a hemispherical dome. **a** Schematic illustration showing the approximation used in the analysis. The truncated and untruncated films are approximated as circular films of radius 3.5 mm ($R_{fs}$) and 9.3 mm ($R_{fl}$), respectively. **b,c** Radial **b** and hoop **c** strain distributions in the two films. **d,e** Radial **d** and hoop **e** strain plotted as a function of the radial coordinate $r$. Highlighted zones indicate tolerable strain levels. **f** Maximum tensile radial strain (left) and tensile hoop strain (right) of films with different radii

unique advantages, such as the superb photo-absorption coefficient ($>5 \times 10^{7}\,m^{-1}$)[25], photoresponsivity (2200 A W$^{-1}$)[26], and high fracture strain ($\sim 23\%$)[23]. The softness[23, 27] and ultrathin thickness[28, 29] of MoS$_2$ are additional factors that enable the fabrication of the soft optoelectronic device. However, an ultrasoft MoS$_2$-based multicell optoelectronic device that can capture images on the hemispherical surface and its application to soft bioelectronics have not been reported yet.

Here, we present an ultrasoft and high-density curved MoS$_2$-graphene photodetector array using single-lens optics. Unique advantages of the soft omnidirectional CurvIS array include the high-density array design (Supplementary Fig. 3b), small optical aberration (Supplementary Figs. 1b, 2), and simplified optics (Supplementary Fig. 1b). The MoS$_2$-graphene-based CurvIS array shows much lower induced strain than the fracture strain of composing materials because of the ultrathin thickness and softness of 2D materials[23, 30]. In addition, the truncated icosahedron design (fullerene-like structure) and the strain-isolation device design enable the CurvIS array to have an almost complete coverage on the hemispherical surface (Supplementary Fig. 4). The high-density MoS$_2$-graphene CurvIS array successfully recognizes various projected images without infrared (IR) noise. It is the first attempt to achieve high-quality imaging using the ultrathin MoS$_2$-based optoelectronic device in a hemispherically curved format with the single-lens optics. Then we propose a human eye-inspired soft implantable optoelectronic device consisting of the CurvIS array and ultrathin neural-interfacing electrodes (UNE) by mimicking structural features of the human eye. A soft and flexible image processing unit is also introduced to construct the fully integrated soft implantable electronic system. The soft CurvIS array and UNE system minimizes mechanical distortion of the retina and effectively stimulates the retinal nerves in response to external optical signals. Detailed theoretical modeling and finite element analysis (FEA) are also performed to understand the mechanics of the proposed materials and device designs in the retina, which highlights importance of the softness and the curved shape of the optoelectronic device in the retinal implant.

## Results

**Materials and device designs for the high-density CurvIS array.** Figure 1a, b show a schematic illustration and a corresponding image of the MoS$_2$-graphene-based high-density CurvIS array. The light signal is focused by the plano-convex lens and measured by the CurvIS array. Constructing a high-density image sensor array on the hemispherical surface (Supplementary Fig. 4) to achieve optical advantages (Supplementary Figs. 1, 2 and Supplementary Note 1) requires development of a novel soft photodetector array. When a conventional film-type image sensor array is laminated on the hemispherical dome, for example, the bending induced strain causes folds and wrinkles in the array that markedly increase the chance of mechanical failures in devices (Supplementary Fig. 5). On the other hand, the high-density CurvIS array can be fabricated on the hemisphere without mechanical fractures (Supplementary Fig. 4 and Fig. 1e) by introducing the ultrathin device structure (51 nm; Fig. 1c, d), using inherently soft materials (MoS$_2$[23] and graphene[30]; Supplementary Fig. 6), applying a strain-isolation device design[31] (isolation of Al$_2$O$_3$ and Si$_3$N$_4$; Fig. 1c, 1c inset, and Supplementary Fig. 7a), and introducing a truncated icosahedron design (Fig. 1a, b). The resulting CurvIS array successfully visualizes the focused optical image (e.g., university logo; inset of Fig. 1b).

The phototransistor array is composed of a MoS$_2$-graphene heterostructure (6 nm; synthesis process in Methods) and other nanomembranes (Al$_2$O$_3$ dielectric (25 nm), Ti/Au gate (5/10 nm), and Si$_3$N$_4$ substrate (5 nm)). As shown in Fig. 1d, the layer number of the MoS$_2$ and graphene are six and four, respectively. The entire thickness of the device is 51 nm (Fig. 1c, d), which is much thinner than conventional silicon-based photodetectors whose thickness is in the range of micrometers or thicker. Top and bottom polyimide (PI) encapsulations (420 nm each) protect the device. The ultrathin thickness of the device dramatically decreases the bending induced strain in the curved system[7, 8]. Furthermore, MoS$_2$ and graphene, which are used as a photo-absorbing layer and interconnection, respectively, have the much higher fracture strain ($\sim 23\%$ and $\sim 25\%$, respectively)[23] than silicon ($\sim 1\%$)[32]. Unlike silicon that needs a thick active layer due

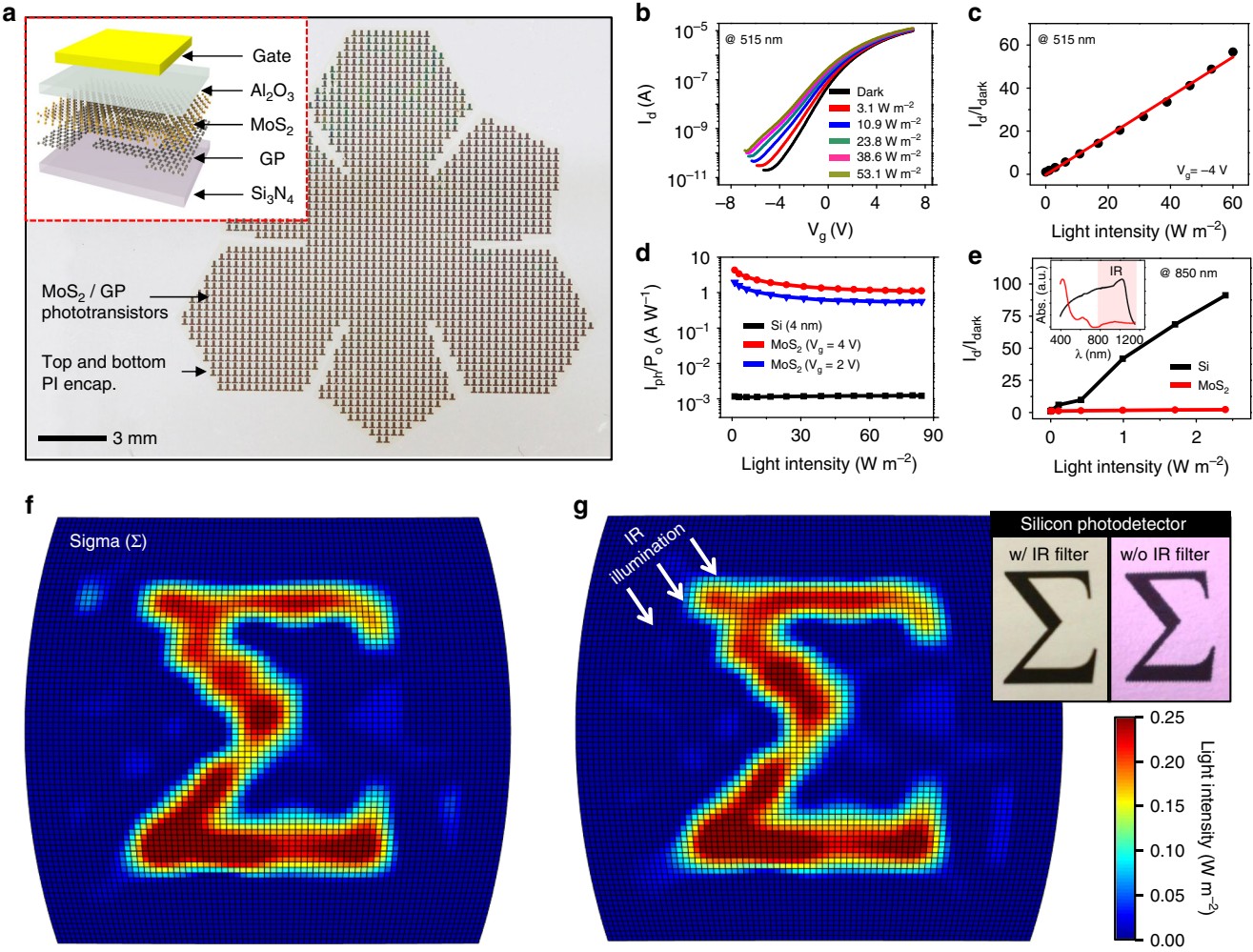

**Fig. 3** Device characterization and imaging using the curved image sensor array. **a** The optical camera image of the phototransistor array with a truncated icosahedron design on a planar substrate. Inset shows a schematic illustration of the device structure. **b** Transfer curves of the phototransistor under different light (515 nm) intensities. **c** Normalized photocurrent change under different light intensities. **d** Photoresponsivity of the MoS$_2$-graphene phototransistor compared to the silicon photodetector with the same thickness. **e** Normalized photocurrent change under IR illumination (850 nm) of different light intensities. Inset shows the light absorbance of MoS$_2$ and silicon. **f** Sigma-shaped image captured by the CurvIS array. **g** The same image with Fig. 3f but captured under IR illumination. Inset images are acquired by a commercial silicon photodetector array with (left) and without (right) an IR filter under IR illumination

to its low photo-absorption coefficient, MoS$_2$ is atomically thin[23] and has a high photo-absorption coefficient[25], both of which are favorable for fabrication of a much thinner photo-absorbing layer.

**Theoretical analysis of the soft optoelectronic device based on mechanics**. Theoretical analyses of a flat membrane fully conforming to a rigid hemispherical surface corroborate the validity of the proposed materials and device designs[33, 34]. When conforming the device array to the hemispherical dome, two types of mechanical failures make the process challenging. First, the large tensile strains can cause direct fracture of constituent materials. Second, the excessive compressive hoop strain near the edge of the film induces buckle delamination and folds in the film[34], which leads to additional fractures due to large local curvatures. For quantitative analysis, we adapt simplified models that the truncated icosahedron design is approximated as seven separate small circular films (Fig. 2a top; $R_{fs} = 3.5$ mm), while the design without cutting is approximated as a large circular film (Fig. 2a bottom; $R_{fl} = 9.3$ mm). Analytical solutions in Supplementary

Note 2 yield the radial and hoop strain distributions for the truncated and untruncated films fully conformed to a hemispherical dome ($R_d = 11.34$ mm), which are plotted as contour plots (Fig. 2b, c, respectively) and curves (Fig. 2d, e, respectively). Comparing with the film without cutting, the overall tensile strain level in the film with the truncated icosahedron design is significantly lower in all regions (Fig. 2d). Figure 2f plots the maximum radial (left) and hoop (right) strains as functions of the normalized film size. It is obvious that both maximum radial and hoop tensile strains increase monotonically with the film size. Without the truncated icosahedron design ($R_{fl}/R_d = 0.82$), maximum radial and hoop strains are as large as 2.86% and 2.77%, respectively. In addition, a significant portion of the flat film is subjected to the tensile strain higher than 1%, which is the fracture strain of inorganic materials (e.g., Al$_2$O$_3$ and Si$_3$N$_4$). With the truncated icosahedron design ($R_{fs}/R_d = 0.31$), in contrast, the maximum radial (0.41%) and hoop (0.40%) strains are well below 1%, which ensures mechanical integrity of all materials in the device. The proposed design also prevents delamination of the CurvIS array from the hemispherical dome. Without the truncated icosahedron design, the high compressive hoop strain

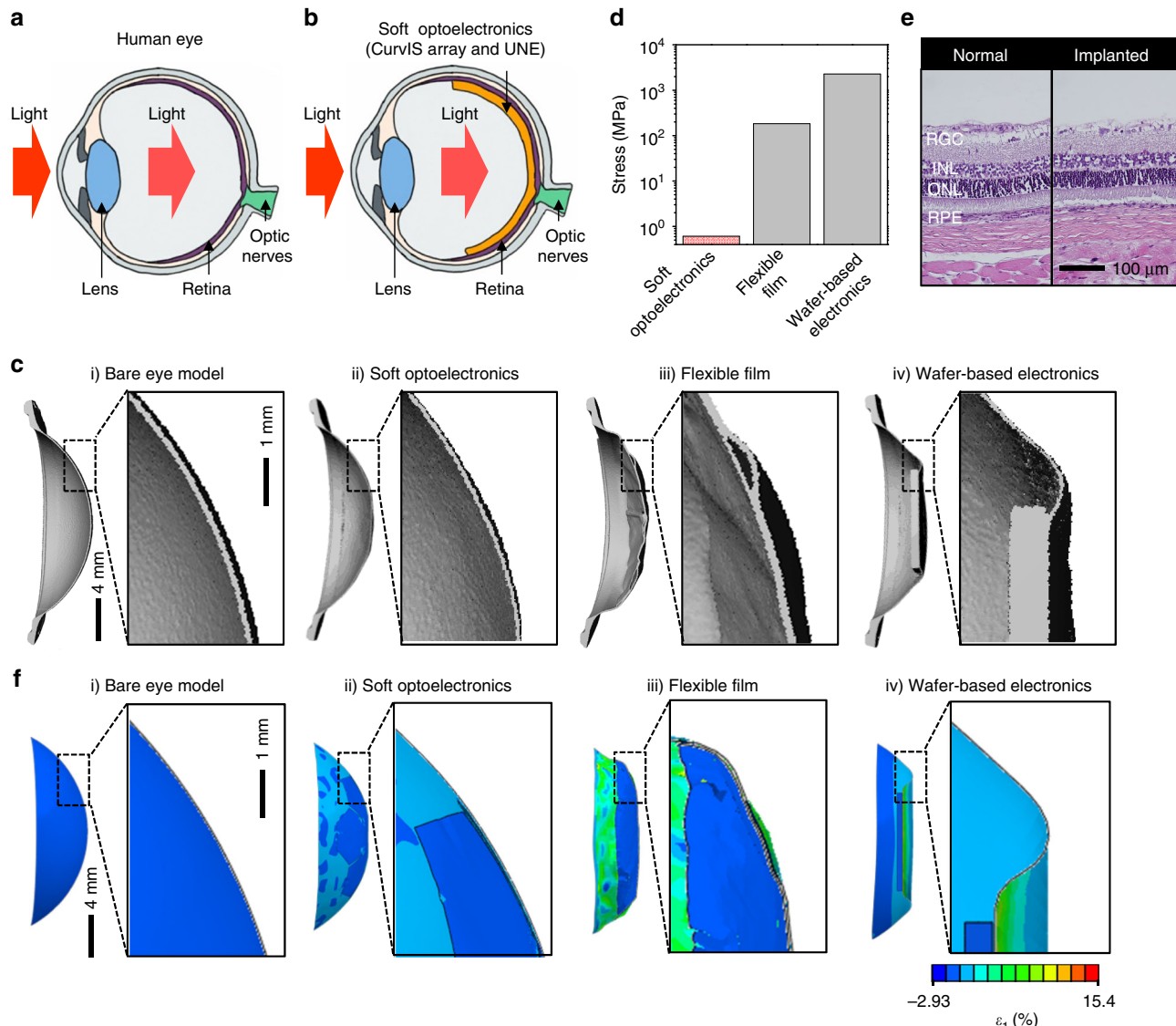

**Fig. 4** Human eye-inspired soft optoelectronic device. **a** Schematic illustration showing the ocular structure of human. **b** Schematic illustration showing the ocular structure with the soft optoelectronic device. **c** Micro CT image (left) and magnified image (right) showing deformation of (i) the bare eye model, attached by (ii) the soft optoelectronic device, (iii) a flexible film device, and (iv) wafer-based electronics. **d** Induced Stress by three different implanted devices. **e** The H&E stain histology of the normal retina and the retina implanted with the soft optoelectronic device. **f** FEA results of the maximum principal strain in eye model (i) without any device, (ii) with the soft optoelectronic device, (iii) with a flexible film device, and (iv) with wafer-based electronics

near the edge of the film (black line in Fig. 2e) can lead to buckle delamination and self-folding in the device film (Supplementary Fig. 5). Such large local curvatures induced by delamination and self-folding cause image distortion as well as mechanical fractures of the constituent materials. With the truncated icosahedron design, however, the negative hoop strain is effectively reduced (red line in Fig. 2e) and buckle delamination is hardly observed (Supplementary Fig. 4).

**CurvIS array based on the MoS₂-graphene heterostructure.** A phototransistor array based on the $MoS_2$-graphene heterostructure with the PI encapsulation is fabricated on a flat substrate (Fig. 3a and Supplementary Fig. 7). This array is transferred to a hemispherical surface for fabricating the CurvIS array (Supplementary Fig. 8). Detailed fabrication steps are described in Methods. The exploded schematic of a single phototransistor is shown in the Fig. 3a inset.

The transfer curve ($I_d$–$V_g$) shows a typical light-sensitive field-effect transistor behavior (Fig. 3b). Under illumination (515 nm), the $MoS_2$ channel generates a photocurrent whose normalized magnitude ($I_d/I_{dark}$) is proportional to the illuminated light intensity (Fig. 3c). The photoresponsivity of the $MoS_2$-graphene phototransistor is compared with the theoretical photoresponsivity of a silicon photodiode whose silicon thickness is same as $MoS_2$ (Fig. 3d; details of the photoresponsivity comparison in Methods). It is found that the former is 2–3 orders higher than the latter, which is due to the efficient photo-absorption of $MoS_2$[25]. Conventional silicon image sensors absorb IR light (850 nm) (Fig. 3e inset), which causes IR noises. On the contrary, the $MoS_2$ photodetector does not absorb the IR spectrum because of its wide bandgap (Fig. 3e). Therefore, an IR filter is unnecessary in the $MoS_2$ device, which helps reduce the thickness and increase the softness of the CurvIS array. As shown in Supplementary Fig. 9, the calibrated $MoS_2$-graphene phototransistor array

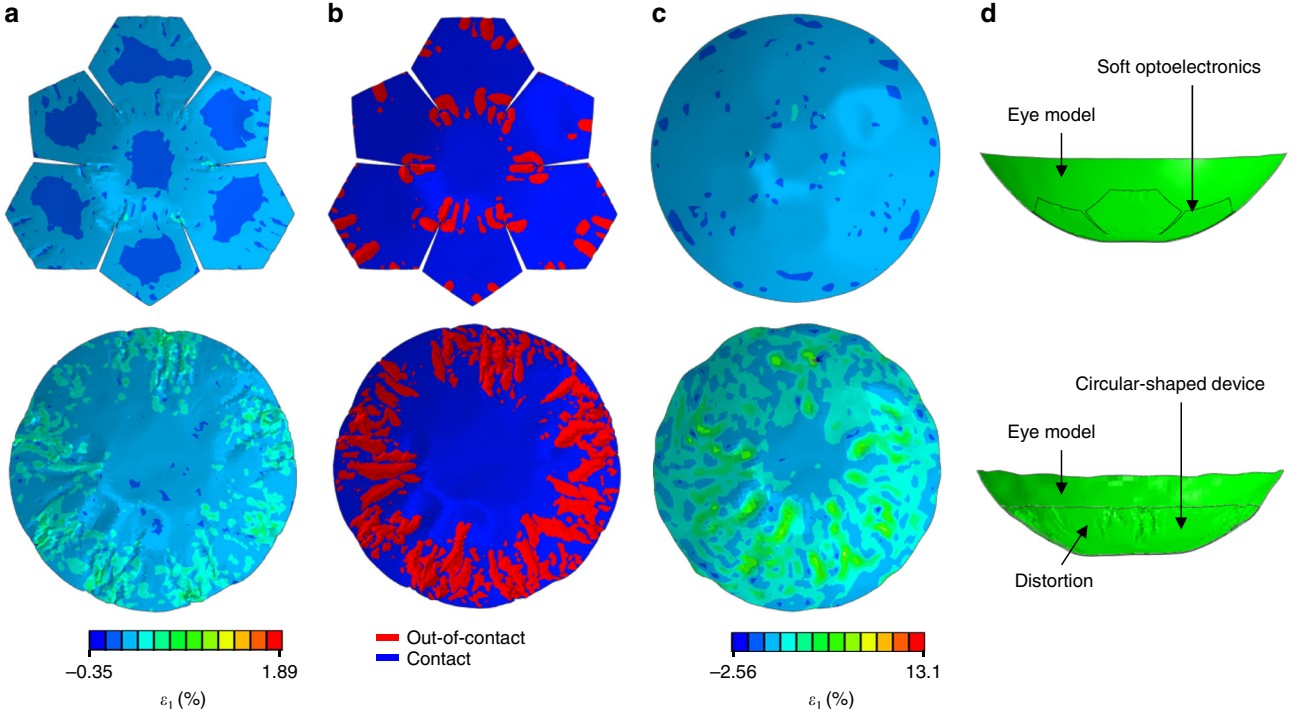

**Fig. 5** Finite element analysis of the soft optoelectronic device and the eye model. **a** Maximum in-plane principle strain distribution in the soft optoelectronic device (top) and the circular device (bottom). **b** The reddish out-of-contact part of the soft optoelectronic device (top) and the circular device (bottom). **c** Maximum in-plane principle strain distribution in the eye models attached by the soft optoelectronic device (top) and the circular device (bottom). **d** Deformed shape of the eye model attached by the soft optoelectronic device (top) and the circular device (bottom) obtained by the FEA

presents the spatially uniform signal distributions under two different light intensities (2.2 W m$^{-2}$ and 3.7 W m$^{-2}$).

The CurvIS array captures various images successfully (Fig. 1b inset, 3f, 3g). The CurvIS array visualizes the alphabet sigma (Σ) (Fig. 3f) using a single plano-convex lens (Supplementary Fig. 1b) installed in a customized setup (Supplementary Fig. 10). Detailed optical imaging procedures are included in Methods. The captured image is not affected by the IR radiation (Fig. 3g) due to IR blindness of the MoS$_2$-based phototransistor (Fig. 3e), whereas a conventional silicon photodetector array without an IR filter shows reddish IR noises (Fig. 3g inset right). Other imaging results (e.g., cross and heart) are shown in Supplementary Fig. 11.

**Human eye-inspired soft implantable optoelectronics**. The ultrathin CurvIS array whose shape and mechanical softness are similar to those of the human retina has high potential to be used as a soft photodetecting component in the retinal prosthesis. Hence, the developed ultrathin CurvIS array is applied to the human eye-inspired soft implantable optoelectronic device. The human eye consists of a lens that collects light, a retina that converts lights into action potentials, and optic nerves that transmit action potentials to the brain (Fig. 4a). As the photo-receptors are distributed over the hemispherical retina, the human eye can recognize exact shapes of objects with a single lens. Patients with retinal degeneration, however, lose vision because the incoming light cannot activate optic nerves due to degenerated photoreceptors[35]. The retinal prosthesis restores the vision by acquiring optical information through image sensors, converting the measured optical information into electric signals, and stimulating optic nerves using an electrode array.

Conventional retinal prostheses consist of an external camera module (e.g., camera on eyeglasses) connected to an intraocular micro-electrode array (Supplementary Fig. 12a)[36]. This bulky wearable camera module is uncomfortable, causes unnatural appearances, and leads to image fading due to absence of efference copy from the eye movement[37]. Recently, retinal prostheses using intraocular image sensors have been reported as alternatives (Supplementary Fig. 12b), but these still suffer from various issues; absence of a multi-lens system for focusing images[37] and unwanted immune responses caused by non-conformal integration and/or mechanical mismatch[15] between soft retina and rigid devices.

As shown in Fig. 4b, we propose a soft implantable optoelectronic device by mimicking the structural features of the human eye. The ultrathin soft optoelectronic device consisting of the CurvIS array and UNE is conformally laminated on the hemispherical retina. This configuration enables the compact optic system, broadens the viewing angle, and captures lights over a large area, just like a human retina (Supplementary Fig. 12c). It is especially important to integrate the optoelectronic device onto the hemispherical retina without retinal deformation. The mechanical mismatch between the implanted device and retina may apply continuous pressures to the eye and cause neural degradation particularly in long-term implantation[1, 2, 8]. This potentially leads to further degeneration of photoreceptors[14] and immune responses[1].

An artificial retina and sclera model (i.e., a double-layered elastomeric hemispherical shell having similar modulus with human eye) is prepared to reveal mechanical deformation of the eye by the device implantation (Supplementary Fig. 13). As shown in Fig. 4c, the soft optoelectronic device conforms to the artificial eye model (i; original model) with minimal deformation (ii; 1.4 μm-thick soft optoelectronic device), while lamination of a flexible film (iii; 15 μm-thick flexible film) and wafer-based electronics (iv; 525 μm-thick silicon device) induce significant distortion. Detailed mechanical analyses regarding the interfacial tractions between three different implantable devices and the eye

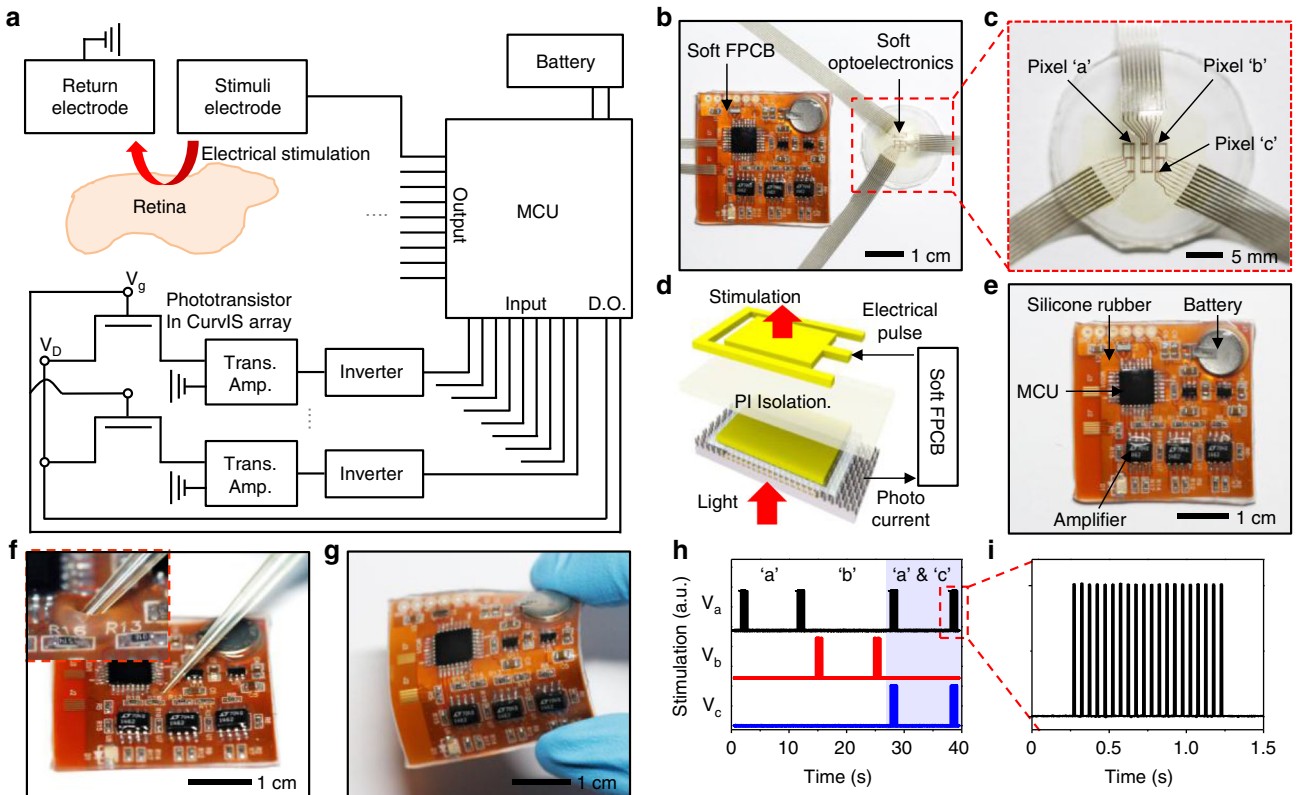

**Fig. 6** Soft flexible printed circuit board that integrates the CurvIS array with UNE. **a** Schematic drawing of the electronics for detecting the external light (bottom) and for applying the stimulation (top). **b** Optical camera image of the CurvIS array and the UNE on the eye model, which are connected by the soft FPCB. **c** Magnified optical camera image of the vertically stacked the CurvIS array and the UNE. **d** Schematic illustration of the phototransistor (bottom) and the stimulation electrode (top) stacked together and connected via the soft FPCB. **e** Optical camera image of the soft FPCB. **f, g** Optical camera image of the soft FPCB under poking **f** and bending **g**. **h, i** Generated electrical pulses at three different pixels by responding the light on/off **h**, and magnified electrical pulse **i**

model are described in Supplementary Note 3. Smaller interfacial traction is a critical factor because the traction deforms the soft eye model. The soft optoelectronic device causes stress to the artificial eye model in the orders-of-magnitude lower level than others (0.61 MPa; Fig. 4d), hence inducing minimal deformation to the eye model (ii; Fig. 4c). The interfacial traction between the flexible film and the eye model is estimated to be 183 MPa, which would induce visible distortion in the eye model (iii; Fig. 4c). When attaching the wafer-based electronics to the eye model, the upper limit of the required traction is 2.27 GPa, which induces significant deformation of the eye model (iv; Fig. 4c).

Minimal mechanical disturbance by the soft optoelectronic device is also analyzed by comparing the histology results of the device-implanted retina (experiment group) and the normal retina (control group). The soft optoelectronic device implanted in the retina both for the short (1 week) and long (9 weeks) period shows good biocompatibility in comparison with the control group (normal retina; Fig. 4e and Supplementary Fig. 14). The expression of the fibroblast growth factor 2 (FGF2)[38] and the glial fibrillary acidic protein (GFAP)[38] in the retina implanted with the soft optoelectronic device show similar tendency with those in the normal retina (Supplementary Fig. 14), which indicates the long-term mechanical and material biocompatibility.

**FEA of the various implantable devices and the eye model.** Although the theoretical analysis in Fig. 2 elaborates the mechanical benefits of the truncated icosahedron design, we

assume that the hemispherical dome is rigid. However, the retina and sclera structure is actually soft and thin. To numerically compare the mechanical deformation of the soft eye model by the implantation of three different types of devices (Fig. 4c), we perform FEA to simulate the strains induced in the eye model as well as the implanted devices. Figure 4f plots the distribution of principle strain in the eye model without any device (i; Fig. 4f) and with the lamination of the soft optoelectronic device (ii; Fig. 4f), the flexible film (iii; Fig. 4f), and the wafer-based electronics (iv; Fig. 4f). While the soft optoelectronic device causes no visible distortion, the flexible film and the wafer-based electronics induce significant shape change to the soft eye model, which is not acceptable in practice. Quantitatively, the maximum strain in the eye model induced by the soft optoelectronic device, the flexible film, and the wafer-based electronics are 1.81%, 15.4%, and 9.68%, respectively.

FEA can also be employed to realistically display the effects of the truncated icosahedron design of the soft optoelectronic device (upper row of Fig. 5 and Supplementary Video 1) in comparison with the untruncated circular-shaped device which has the same thickness and materials with our soft optoelectronic device (lower row of Fig. 5 and Supplementary Video 2). Figure 5a shows the strain distribution in the devices. The maximum strain in the soft optoelectronic device is limited to 0.4% after conformal lamination, whereas that in the circular device can be up to 1.89%, which can cause the fracture of the comprising materials. In addition, much fewer wrinkles are observed in the soft optoelectronic device than the untruncated circular film in both experiments and FEA (Figs. 4c and 5b, respectively). The reddish

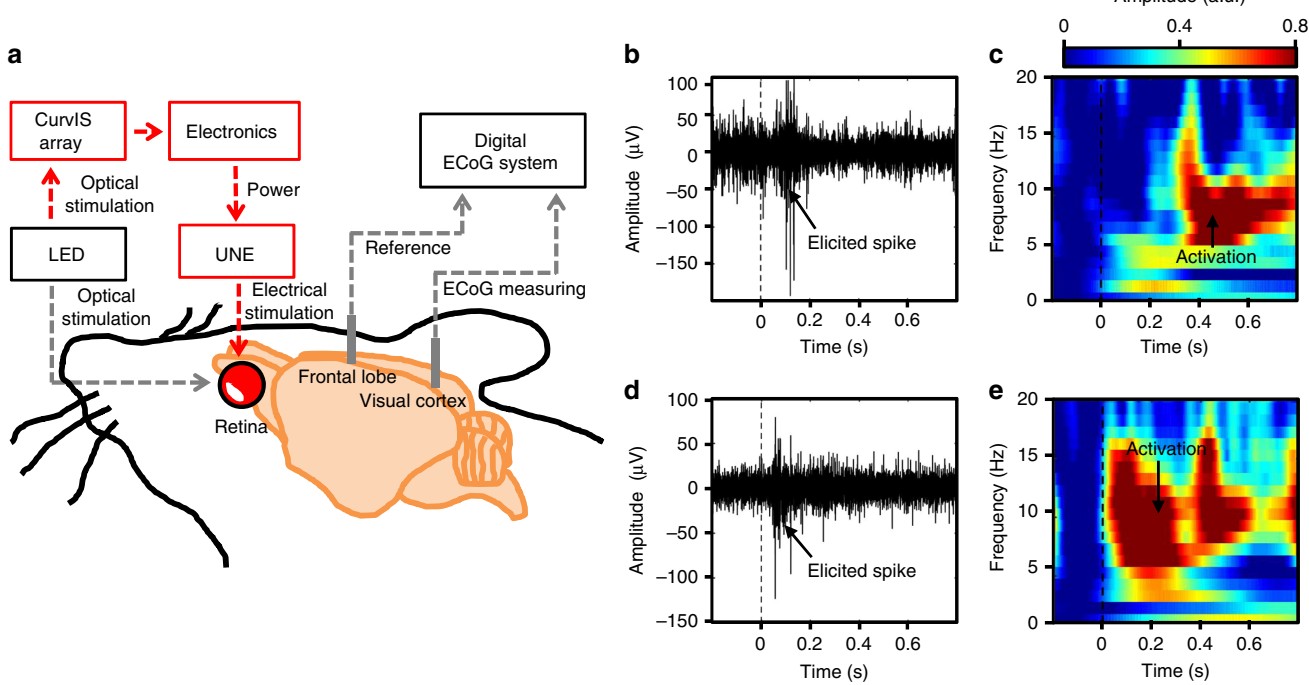

**Fig. 7** Retinal stimulation by the soft optoelectronic device. **a** Schematic drawing of the experimental setup for stimulating the retina (left) and for recording neural signals from the visual cortex (right). **b**, **c** Measurement of elicited spikes **b** and LFP changes **c** in the visual cortex by optical stimulation. **d**, **e** Measurement of elicited spikes **d** and LFP changes **e** in the visual cortex by electrical stimulation

out-of-contact area means the detachment of the device where the gap between the device and the eye model is beyond the thickness of the device (Fig. 5b). This wrinkle-induced delamination can significantly diminish the conformability of the device on the eye model. On the other hand, Fig. 5c offers a striking contrast between the maximum strains in the eye model. The strain in the eye model induced by the soft optoelectronic device is only up to 1.81%, whereas the eye model can be deformed to a maximum strain of 13.1% by the circular device. Such distortion is further visible by cross-sectional views (Fig. 5d). In addition, the 15 μm-thick flexible film (iii; Fig. 4c, f) and the 525 μm-thick wafer-based electronics (iv; Fig. 4c, f) can induce much larger strain to the eye model because of its larger thickness and rigidity. Therefore, we can conclude that the soft optoelectronic device can significantly improve the conformability to the eye model, diminish the strain induced in the device, and reduce the distortion of the soft eye model.

**Flexible electronic system integrating the CurvIS array and UNE**. One of the important issues in the retinal implant is how to convert the visual information obtained by the image sensor array to the corresponding electrical signals to be conveyed to the retina via the micro-electrode array[39]. In commercial retinal implants[40] (e.g., Argus II, Second Sight), the visual information is recognized by a wearable camera module and translated to the electric signals by a video processor to be transmitted to the intraocular micro-electrode array. The electronic devices that supply power and control the system are usually implanted in the extraocular position due to the spatial limitation, but these rigid and bulky devices may cause immune responses and mechanical damages to the surrounding tissues. A photovoltaic type retinal prosthesis without external power sources has been recently reported[14, 41], but head-mounted glasses are still needed to transfer the IR beam to Si photovoltaic devices. Therefore, the soft CurvIS array and UNE integrated with the flexible implantable electronics[42] coated with the soft silicone rubber can be a promising candidate of the

soft retinal implant due to minimal mechanical mismatch[15] between the tissue and the implanted device.

Figure 6a, b show a schematic and a corresponding optical camera image of the integrated soft electronic system. The photocurrent is generated by each phototransistor of the soft CurvIS array in response to external light, and is amplified by a transimpedance amplifier and an inverter. The micro-controller unit measures the amplified signal, processes it, and produces programmed electrical pulses. The pulse electrically stimulates the retina via the electrode stacked with the corresponding photo-transistor. The ultrathin soft optoelectronic device array is conformally laminated on the eye model (Fig. 6c). As shown in Fig. 6d, each phototransistor of the CurvIS array and a corresponding electrode in the UNE are vertically stacked in the ultrathin and soft platform. To develop a soft form of the fully implantable system, we introduce the flexible printed circuit board with the soft surface coating (soft FPCB; Fig. 6e). The soft FPCB includes all electronics for image processing as depicted in Fig. 6a (also see Supplementary Fig. 15 and Supplementary Table 4), analyzes the photocurrent produced from the photo-transistor, and transfers the programmed electrical pulses to the stimulation electrode integrated in the same pixel (Fig. 6d).

Mechanical flexibility and softness of the soft FPCB is confirmed by experimental analyses. The conventional rigid electronics has the modulus in the range of GPa and shows significant mechanical mismatch to the soft human tissues. The surface of the FPCB is coated with thick silicone rubber whose modulus (~ 50 kPa) is similar to that of human tissues (100–1500 kPa)[1]. This mechanically matched material property allows soft and conformal interfaces with surrounding tissues. Figure 6f shows the soft FPCB poked by the tip of a pipette tube. Thick coating of the FPCB with silicone rubber provides the cushion-like surface. The silicone rubber coating also effectively protects the electronic chips from external impact and water exposure. Unlike conventional rigid electronics, the soft FPCB also can be easily deformed (Fig. 6g).

The integrated form of the soft optoelectronic system can successfully recognize the illuminated light and generate programmed electrical pulses. The soft integrated system with the FPCB measures the photocurrent generated at each phototransistor, and delivers electrical stimulation to the eye model using the integrated electrode. When light is illuminated to the pixel 'a' and pixel 'b' as shown in Fig. 6c, electrical pulses are selectively generated at the electrode of the pixel 'a' and pixel 'b', respectively (Fig. 6h white region). Figure 6i shows the magnified electrical signals in Fig. 6h. When light is simultaneously illuminated to the pixel 'a' and pixel 'c', the soft FPCB processes the measured signals from image sensors and successfully generates electrical pulses on both electrodes of pixel 'a' and pixel 'c' (Fig. 6h blue region).

**Retinal stimulation by the soft optoelectronic device.** Figure 7a shows a schematic that describes the in vivo animal experiment using the soft optoelectronic device and a neural recording system. The extraocular imager is used to detect the incoming light signals without causing interferences with healthy photoreceptors. The electrode attached on the retina successfully stimulates the optic nerves (Supplementary Fig. 16a, 16b). The excitation of optic nerves is monitored by penetrative electrodes at the visual cortex (Fig. 7a and Supplementary Fig. 16c). The stimulation of the retina[4, 14] is confirmed by the elicited spikes[43] and changes in local field potential (LFP)[3], which are simultaneously measured at the rat's primary visual cortex[12, 43]. Detailed animal preparation and experiment conditions are described in Methods. When pulsed optical signals are applied to the rat's eye, the rat's retina senses the light being on/off. This optical information is transferred to the visual cortex via optic nerves, resulting in elicited spikes (Fig. 7b) and LFP changes in the frequency range of 4–15 Hz (Fig. 7c). Similarly, the $MoS_2$-graphene-based soft optoelectronic device detects the light being on/off, and the corresponding electrical pulses are applied to the optic nerve through the UNE. Consequently, the visual cortex is similarly activated, leading to elicited spikes (Fig. 7d) and LFP changes in the same frequency range (Fig. 7e).

## Discussion

We could capture various images using the ultrathin high-density CurvIS array and single-lens optics, and demonstrate the prototype of the soft retinal implant consisting of the CurvIS array, UNE, and soft FPCB. To improve the imaging quality of the soft retinal implant, the large amount of optical information obtained by the large number of image sensors should be effectively processed and transferred to the corresponding stimulation electrodes[39]. However, the increase in the pixel density is challenging because the number of interconnecting wires is proportional to the number of pixels[39]. The miniaturization and integration of the retina prosthesis depends on improving the connectivity between the high-density sensor matrix and electronics. Alpha-IMS[37], one of the leading retinal prostheses, can be a good approach to solve this issue. Alpha-IMS[37] has 1,500 pixels, each of which contains a photodetector, an integrated circuit, and an electrode. The electronic circuit in the individual pixel processes and generates the electrical pulses by itself, and hence it minimizes the number of external wires and provides the improved connectivity between high-density matrix of sensors and electronics. The similar strategy can be applied to the soft retinal implant by fabricating the self-processable pixel array composed of the ultrathin photodetector and electrode pair in the future. Innovation in the circuit design (e.g., application specific integrated circuit) can also effectively miniaturize the size of the electronic components for the retinal implant.

In summary, the high-density $MoS_2$-graphene CurvIS array is developed by using ultrathin soft materials and strain-isolating/-releasing device designs. Mechanical and optical analyses corroborate the validity of proposed materials and device designs for the CurvIS array. The CurvIS array with the single-lens optics effectively obtains pixelated images without IR noises. The CurvIS array and UNE are integrated through the soft FPCB to form a human eye-inspired soft implantable optoelectronic device, which causes minimal mechanical deformation to the eye model as validated by both experiments and corresponding FEA simulations. The soft optoelectronic device successfully stimulates the optic nerves of a rat model in response to the pulsed external light, which is confirmed by recording spikes and LFP changes at the visual cortex of the rat. The proposed human eye-inspired soft optoelectronic device is a step forward to the next-generation soft bioelectronics and the soft imaging element of the retinal prosthesis.

## Methods

**Synthesis of the ultrathin $MoS_2$ film.** An ultrathin $MoS_2$ film was synthesized on a $SiO_2$ wafer using chemical vapor deposition (CVD). Crucibles containing 0.1 g sulfur (Alfa Aesar, USA) and 0.3 g $MoO_3$ (Sigma Aldrich, USA) were placed at the upstream and center of the chamber, respectively. The growth substrate was treated using the piranha solution and oxygen plasma, and placed at the downstream of the $MoO_3$ crucible. The chamber was annealed at 150 °C before the synthesis. The chamber was heated at a rate of 25 °C min$^{-1}$, and maintained at 650 °C and 10 torr with Ar flow of 50 sccm for 5 min. The temperature of sulfur was maintained at 160 °C during the reaction. The chamber was naturally cooled after the reaction was complete. A Raman spectrum of the synthesized $MoS_2$ film was analyzed using T64000 (Horiba, Japan) at NCIRF (Supplementary Fig. 6a).

**Synthesis of the ultrathin graphene film.** A graphene film was synthesized using the CVD. A 25 nm-thick copper foil (Alfa Aesar) was cleaned with isopropyl alcohol and annealed at 1000 °C for 30 min under constant $H_2$ flow (8 sccm, 0.08 Torr). After 30 min annealing, additional $CH_4$ flow (20 sccm, 1.6 Torr) was introduced for 20 min at 1000 °C. When the synthetic procedure was finished, the chamber was rapidly cooled to room temperature under $H_2$ flow (8 sccm, 0.08 Torr). The synthesized graphene film is analyzed by Raman spectroscopy (Supplementary Fig. 6b).

**Fabrication and characterization of the CurvIS array.** The fabrication process of the CurvIS array began with spin-coating of a thin PI film (420 nm, bottom encapsulation; Sigma Aldrich) on a $SiO_2$ wafer. A thin layer of $Si_3N_4$ (5 nm, substrate) was deposited using the plasma-enhanced CVD. Using photolithography and dry etching, an island-shape array of the $Si_3N_4$ film was defined. The graphene layer (2 nm) was transferred onto the $Si_3N_4$ layer and patterned as an interdigitated source/drain electrode whose channel length is 10 μm. The ultrathin $MoS_2$ layer (4 nm, photo-absorbing layer) was transferred onto the graphene electrodes, and patterned by photolithography and reactive ion etching. Ti/Au layer (5/10 nm) was deposited for the etch mask and used as probing pads. The $Al_2O_3$ dielectric layer (25 nm) was deposited at 200 °C through the thermal atomic layer deposition. The $Al_2O_3$ layer was etched with a buffered oxide etchant after photolithography as an island-shape array. Then lift-off process was used to pattern the Ti/Au layer (5/10 nm, gate electrode) deposited by the thermal evaporation. Additional spin-coating of a thin PI film (420 nm, top encapsulation) and dry etching completed fabrication of the phototransistor array with a truncated icosahedron design. The truncated icosahedron structure is an Archimedean solid made by cutting out the corners of a regular twenty-sided face and composed of 12 pentagons and 20 hexagons (Supplementary Fig. 17). Its shape is similar with the Telstar soccer ball and fullerene ($C_{60}$). We utilized the partial structure of the truncated icosahedron design to minimize the induced mechanical stress on the hemispherical surface. The fabricated phototransistor array was detached from the $SiO_2$ wafer with a water-soluble tape (3 M Corp., USA). This tape was cut into the truncated icosahedron design. The detached phototransistor array was transfer-printed onto the polydimethylsiloxane (PDMS; Dow, Corning, USA) hemispherical dome (Supplementary Fig. 8).

**Comparison of photoresponsivity between the $MoS_2$-graphene phototransistor and the silicon photodiode with the same thickness.** The silicon photodiode consists of single crystal silicon 1.25 μm-thick PN junction. The fabrication of the film-type silicon photodiode array began with spin-coating of the precursor solution of the thin PI film (420 nm) on a $SiO_2$ wafer. 1.25 μm-thick Si nanomembranes were prepared from the silicon-on-insulator wafer (SOITEC, France), which was doped by boron and phosphorous in advance, and then transferred to the prepared PI film. Each photodiode pixel was fabricated by

photolithography and dry etching. Additional spin-coating of a thin PI film (420 nm, top encapsulation) and following photolithography and dry etching completed fabrication of the film-type device array. Since the photoresponsivity of silicon is proportional to the thickness[44], the theoretical photoresponsivity of 3 nm-thick silicon photodiode was calculated by dividing the photoresponsivity of 1.25 μm-thick silicon device by the thickness. The estimated photoresponsivity of the silicon device was compared to that of the $MoS_2$-graphene-based phototransistor (Fig. 3d).

**Customized imaging setup for the CurvIS array**. The detailed illustration and optical camera image for the imaging setup are shown in Supplementary Fig. 10. A white light-emitting diode (LED; Advanced Illumination, USA) blocked with a metal shadow mask generated the patterned light, the aperture controlled the quantity of the passed light, and the plano-convex lens focused the light on the CurvIS array. The CurvIS array was constructed on a transparent convex hemispherical dome composed of PDMS. Since the patterned light is illuminated from the bottom side, passed through the transparent PDMS support of the convex shape, and then reached the CurvIS array on the convex hemisphere can be considered as a concavely curved imager. The calibrated $12 \times 12$ phototransistor array located at the center of the CurvIS array (Supplementary Fig. 9a right) acquired the focused light pattern. The light intensity incident on each pixel was individually measured by probing tips with a parameter analyser (B1500A; Agilent, USA), and then the pixelated image was processed by applying the interpolation function of Matlab (MathWorks, USA). The averaged values of neighboring cells were used for dead pixels. All the captured images were rendered on a concave hemisphere.

**Characterization of mechanical deformation of the eye model by the implanted devices**. To mimic the structure of retina and sclera, an eye model, a double-layered hemispherical PDMS shell, was fabricated; PDMS layers with different modulus (40 kPa and 1.2 MPa) were sequentially coated on the hemispherical mould, cured, and detached from the mould. Three kinds of implantable devices (i.e., soft optoelectronics, flexible film-type device, and wafer-based electronics) were attached onto the concave surface of the eye model, and three-dimensional deformations of the eye models were observed by a micro computed tomography (micro CT, Viva CT 80; Scanco Medical, Swiss).

**FEA of the soft optoelectronic device and the eye model**. The process of attaching the device to the eye model was simulated by a commercial software (ABAQUS). In experiments, the optoelectronic device was attached to the eye model by water evaporation, whereas the flexible film and the silicon wafer were attached by finger. To simulate the integration process in FEA, the devices were conformed to the eye model by an externally applied pressure, which was converted from the surface tension of the water by the Young-Laplace equation or from the pressure applied by finger. After the device attached to the eye model, the externally applied pressure was unloaded. The tangential interaction between the device and the eye model was frictional and the friction coefficient was set to be 0.05, while the normal interaction was no separation after contact. Four-node shell elements were used to model the optoelectronic device, the flexible film, and the eye model. Eight-node solid elements were used to model the silicon wafer. The optoelectronic device was assumed to be a 1.4 μm-thick PI ($E_{PI} = 2.5$ GPa, $\nu_{PI} = 0.34$) because of the ultrathin thickness of the soft optoelectronic device, while the flexible film and silicon wafer were set to be a 15 μm-thick Al film ($E_{Al} = 69$ GPa, $\nu_{Al} = 0.34$) and 525 μm-thick Si wafer ($E_{Si} = 165$ GPa, $\nu_{Si} = 0.22$), respectively. The artificial eye model was modeled to be a bilayer structure consistent with the experiment, i.e., 31 μm-thick softer PDMS (neo-Hookean material, $C_{10,PDMS1} = 6.55$ kPa, $D_{1,PDMS1} = 0.0122$ kPa$^{-1}$) to be the inner layer and 65 μm-thick stiffer PDMS ($C_{10,PDMS2} = 204$ kPa, $D_{1,PDMS2} = 0.0508$ MPa$^{-1}$) to be the outer layer.

**Fabrication of the UNE**. The UNE fabrication began with spin-coating of the PI film (420 nm) on a $SiO_2$ wafer. A lift-off process was used to pattern Cr/Au layer (7/40 nm) and Pt layer (25 nm) deposited by thermal evaporation and sputtering, respectively. Additional spin-coating of PI film (420 nm) and dry etching completed fabrication of the UNE. The electrode of the low impedance (1.31 kΩ at 1 kHz; Supplementary Fig. 18) was used for the neural stimulation.

**Animal preparation for the in vivo experiment**. In this study, we used male Wistar rats whose weights are in the range of 280–300 g and in the age of 10–12 weeks (Japan SLC; Hamamatsu, Japan). The animals were housed at the temperature of 22–24 °C with a 12/12 h light/dark cycle. The rats were given at least 1 week to adapt to their environment before experiments. The Institutional Animal Care and Use Committee at the Korea Basic Science Institute (KBSI-AEC 1601) reviewed and approved this study. All animal procedures were in accordance with the Guide for the Care and Use of Laboratory Animals issued by the Laboratory Animal Resources Commission of KBSI.

**In vivo animal experiment to confirm biocompatibility of the soft optoelectronic device**. The rat's eye in which soft optoelectronic device is implanted for 7 days was compared with the normal eye. The soft optoelectronic device is implanted into the eye by minimally invasive surgery. Antibiotics and dexamethasone were treated to prevent inflammation by the surgical procedures. The

eyes were fixed at 4% paraformaldehyde solution, were embedded in paraffin, and were sliced at coronal plane to 5 μm thickness by a microtome. These sliced tissues were mounted on slide glasses, and stained with Hematoxylin–Eosin (MHS16, HT110180, Sigma Aldrich) by following the standard histochemical procedures. We also utilized 4',6-diamidino-2-phenylindole (Vectors Laboratories, USA), FGF2 (1:200, Santa Cruz biotechnology, USA), and GFAP (1:800, Santa Cruz biotechnology) staining by following the standard protocol to obtain fluorescence imaging data of key factors related with retinal biocompatibility. Histofluorescence images were obtained by a confocal microscope (LSM 780 NLO, Carl Zeiss, Germany).

**In vivo animal experiment for the retinal stimulation**. This study compared optical and electrical stimulation to retina of a healthy rat. The detailed experimental setup is shown in Supplementary Fig. 16. The rat was fixed in a stereotaxic frame. A commercial white LED was fixed in front of the eyeball and delivered light for the optical stimulation. Lensectomy was performed to conformally attach the UNE onto the retina. The extraocular light detection was performed to prevent the activation of the healthy retina and the signal interference by external light. $MoS_2$-graphene-based phototransistors generated photocurrent by the illuminated light, and the photocurrent was amplified by the external amplifier. The data acquisition system detected the amplified photocurrent and triggered a function generator. The function generator applied biphasic electrical pulses (80 μA, 50 μs, and 20 Hz) to the retinal nerves via a single-channel UNE. The optical and electrical stimulations were repeated at least 20 times for statistical analyses. To record the neural responses by the stimulations, Parylene-C insulated tungsten microelectrodes (~ 1 MΩ at 1 kHz, 100 μm diameter) were inserted into the primary visual cortex (7.0 mm posterior to bregma, 3.0–4.0 mm lateral to the midline, 800–1000 μm ventral to dura mater) and frontal lobe (Supplementary Fig. 16c). The neural signal was filtered between 0.3 and 10 kHz and sampled at 25 kHz. The neural signals were separated into spikes and LFPs by 300 Hz high- and 100 Hz low-pass filter, respectively. To remove the stimulation artefacts and detect spikes, the curve-fitting method was used. Time-frequency analysis was performed to allow tracking of the time-varying energy in the frequency band of the LFP signals. To visualize the time-varying energy, we calculated the spectrograms that were baseline-corrected and averaged across all trials.

**Data availability**. The data files that support the findings of this study are available from the corresponding author on reasonable request.

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

## Acknowledgments

This research was supported by IBS-R006-A1 and US NSF CMMI-1541684.

## Author Contributions

C.C., M.K.C., S.L., K.W.C., N.L. and D.-H.K. designed the experiments, analyzed the data and wrote the paper. C.C., M.K.C., M.S.K., M.K., E.J., J.L. and D.S. performed characterization of individual devices. S.L., X. Q. and N.L. performed theoretical analysis on mechanics. G.J.L. and Y.M.S. performed theoretical analysis on optics. C.C., M.K.C., O.K. P., C.I., J.K., S.-H.K. and N.L.J. designed the in vivo animal experiments. All authors discussed the results and commented on the manuscript.

## Additional information

**Competing interests:** The authors declare no competing financial interests.

