## [Peer Review File · Nature Communications]

Reviewers' comments:

Reviewer #1 (Remarks to the Author):

In this manuscript, Choi et al present an impressive and thorough study of a new curved image sensor array that can function as a retinal implant. Overall, the performance and results of the sensor array particularly in in vivo tests, as well as the detailed mechanical analysis and optoelectronic testing are all very well done. I recommend publication after minor revisions noted below:

- The geometry of the truncated icosahedron can be explained a bit more.
- How were the film-type silicon devices fabricated or obtained for comparison to the new ultrathin devices?
- What are the estimated layer numbers of MoS₂ and graphene? (the methods state the thickness in nm, but an estimate of layer numbers would be helpful)
- How are contacts made to the device pixels during imaging testing? In SI Fig 10 it looks like probe tips are used to test each individual pixel sequentially? That should be stated clearly.
- How are the UNE electrodes aligned with the CurVIS devices?
- The motivation to develop retinal implants for particular diseases of the eye could be discussed earlier in the introduction.
- Some awkward phrasing in places (e.g. "thick thickness")

Reviewer #2 (Remarks to the Author):

This work discusses the design and fabrication of photosensitive curved image sensor arrays based on MoS₂-graphene heterostructures.

The main novelty of this manuscript by Choi and colleagues concerns the use of MoS₂-graphene heterostructures to fabricate curved image sensors. However, despite this interesting application (curved imaged sensors), this work does not provide sufficient arguments (in terms of novelty) to justify its publication in Nature Communications. For instance, the work fails to demonstrate an improvement of the photodetector performance in comparison with published work using the same material platform. Further, this work provides a too basic proof-of-concept for implantable optoelectronic device. In this respect, the work fails to demonstrate sufficient level of integration of the photodetector and the stimulation element (no implantable design or device is presented). I am listing below specific comments that could help the authors to improve their work

1. The soft implantable optoelectronic device has not sufficient level of integration of the photosensitive element and the stimulating element. Given the level of advancement in the field (see work by Benfenati in Nature Nanotech and Palanker in Nature Photonics), the device should integrate the photodetector pixels with the stimulating microelectrode pixels. In their experiment, the authors use extraocular detection, and amplifier, and a very basic stimulating device. A real advancement in the field would require integration of these functionalities (as already provided in the state of the art).

2. The authors have chosen to use healthy animals for the in vivo validation of the technology. This referee questions this choice because it is difficult to derive conclusions from the electrical stimulation study.

Reviewer #3 (Remarks to the Author):

The paper entitled "Human-eye-inspired soft optoelectronic device using high-density MoS₂-graphene curved image sensor array" shows a very interesting technological development to improve the performance of the retinal implants. The presented development could improve the sensing capabilities of the current retinal implants, making it more suitable for the patient and

reducing the external components such as cameras or lenses. However, the authors do not take into account the integration of this development in a functional retinal implant. The authors should consider the following points to understand how this technology can be integrated in a retinal implant.

- * The main limitation of the retinal implants is in how the optical information recaptured by an external camera is transferred efficiently to the nervous system. This means, how to increase the number of stimulation sites and how to make this stimulation more effective (in terms of qualitative recognition by the patient)

- * The presented development is just a matrix of photo-detectors without any stimulation capability. The in vivo experiment for retinal stimulation is performed with just one stimulation electrode. The authors should discuss and show how the large amount of information recaptured by the high-density image sensor can be transmitted to the nervous system. At least, the authors should discuss in how to integrate in the sensor a similar range of stimulation electrodes.

- * The authors should also detail the electronics required to read the matrix of photo-detectors, in order to discuss about the complexity of integrating it in a retinal implant (i.e. size and number of required connections).

Reviewer #1:

Summary Comments: *In this manuscript, Choi et al present an impressive and thorough study of a new curve image sensor array that can function as a retinal implant. Overall, the performance and results of the sensor array particularly in in vivo tests, as well as the detailed mechanical analysis and optoelectronic testing are all very well done. I recommend publication after minor revisions noted below:*

Our response to summary comments: We thank the reviewer for the detailed and insightful evaluation on our work.

Comment #1: *The geometry of the truncated icosahedron can be explained a bit more.*

Our response to comment #1: We thank the reviewer for the valuable comment. The truncated icosahedron design is important for the high-density curved image sensor array. We included more detailed explanation of the truncated icosahedron geometry in the revised manuscript.

Our modification to the manuscript:
(Line 9, page 4: in revised main text)

“In addition, the truncated icosahedron design (fullerene-like structure) and the strain-isolation device design enable the CurvIS array to have an almost complete coverage on the hemispherical surface (Supplementary Fig. 4).”

(Line 5, page 18: in revised main text)

“The truncated icosahedron structure is an Archimedean solid made by cutting out the corners of a regular twenty-sided face and composed of 12 pentagons and 20 hexagons (Supplementary Fig. 17). Its shape is similar with the Telstar soccer ball and fullerene (C₆₀). We utilized the partial structure of the truncated icosahedron design to minimize the induced mechanical stress on the hemispherical surface.”

(Supplementary Figure 17: in the revised manuscript)

Supplementary Figure 17 | Truncated icosahedron design. The truncated icosahedron device array design inspired by the truncated icosahedron structure.

Comment #2: How were the film-type silicon devices fabricated or obtained for comparison to the new ultrathin devices?

Our response to comment #2: We appreciate the reviewer’s comment. As an answer to the question from the reviewer, we added the detailed method for fabricating the film-type silicon device.

Our modification to the manuscript:

(Line 18, page 18: in revised main text)

“The fabrication of the film-type silicon photodiode array began with spin-coating of the precursor solution of the thin PI film (420 nm) on a SiO₂ wafer. 1.25 μm-thick Si nano-membranes were prepared from the silicon-on-insulator wafer (SOITEC, France), which was doped by boron and phosphorous in advance, and then transferred to the prepared PI film. Each photodiode pixel was fabricated by photolithography and dry etching. Additional spin-coating of a thin PI film (420 nm, top encapsulation) and following photolithography and dry etching completed fabrication of the film-type device array.”

Comment #3: What are the estimated layer numbers of MoS₂ and graphene? (the methods state the thickness in nm, but an estimate of layer numbers would be helpful)

Our response to comment #3: We appreciate the reviewer’s comment. The estimated layer number of MoS₂ and graphene are four and six, respectively, and we included this information in the revised manuscript.

Our modification to the manuscript:

(Line 20, page 5: in revised main text)

“As shown in Fig. 1d, the layer number of the MoS₂ and graphene are six and four, respectively.”

(Figure 1d: in the revised manuscript)

Figure 1d | Cross-sectional transmission electron microscope image of the MoS₂-graphene phototransistor (left) and the magnified image of the MoS₂-graphene heterostructure (right).

Comment #4: How are contacts made to the device pixels during imaging testing? In SI Fig 10 it looks like probe tips are used to test each individual pixel sequentially? That should be stated clearly.

Our response to comment #4: We thank the reviewer for the detailed comment. We measured the photo-detecting pixels one by one by probing them sequentially. As pointed out from the reviewer, we modified the method to explain it clearly.

Our modification to the manuscript:
(Line 17, page 19: in revised main text)

“The light intensity incident on each pixel was individually measured by probing tips with a parameter analyser.”

Comment #5: How are the UNE electrodes aligned with the CurVIS devices?

Our response to comment #5: We appreciate the reviewer’s constructive comment. The phototransistor and stimulation electrode are located at the same position but are electrically isolated by the polyimide passivation layer. They are electrically connected through external wiring and electronics on a flexible printed circuit board (FPCB) coated with ultrasoft silicone rubber. We added the detailed explanations in Fig. 6.

Our modification to the manuscript:
(Figure 6: in the revised manuscript)

Figure 6 | Soft flexible printed circuit board that integrates the CurVIS array with UNE. (a) Schematic drawing of the electronics for detecting the external light (bottom) and for applying the stimulation (top). **(b)** Optical camera image of the CurVIS array and the UNE on the eye model, which are connected by the soft FPCB. **(c)** Magnified optical camera image of the vertically stacked the CurVIS array and the UNE. **(d)** Schematic illustration of the phototransistor

(bottom) and the stimulation electrode (top) stacked together and connected via the soft FPCB. (e) Optical camera image of the soft FPCB.

Comment #6: The motivation to develop retinal implants for particular diseases of the eye could be discussed earlier in the introduction.

Our response to comment #6: We appreciate the reviewer's valuable comment. We modified the introduction to address the comment from the reviewer.

Our modification to the manuscript:
(Line 23, page 2: in revised main text)

“Likewise, the soft bioelectronic device can play an important role in the intraocular retinal prosthesis for patients with retinal degeneration (e.g., macular degeneration or retinitis pigmentosa). As the optic nerves widely spread in the soft (~ 20 kPa)¹³ and hemispherically-shaped retina, a soft and curved form of the high-density image sensor and electrode array which mechanically matches with the human retina is significantly needed particularly for the long-term retinal prosthesis.”

Comment #7: Some awkward phrasing in places (e.g. "thick thickness")

Our response to comment #7: We appreciate the reviewer's detailed comment. We modified the awkward phrasing as follows.

Our modification to the manuscript:
(Line 4, page 6: in revised main text)

“thick thickness” → “**thick active layer**”

Thank you very much again for your insightful comments. We feel that these comments have helped to improve the quality of the manuscript significantly.

Reviewer #2:

Summary Comments: *This work discusses the design and fabrication of photosensitive curved image sensor arrays based on MoS₂-graphene heterostructures. The main novelty of this manuscript by Choi and colleagues concerns the use of MoS₂-graphene heterostructures to fabricate curved image sensors. However, despite this interesting application (curved imaged sensors), this work does not provide sufficient arguments (in terms of novelty) to justify its publication in Nature Communications. For instance, the work fails to demonstrate an improvement of the photodetector performance in comparison with published work using the same material platform. Further, this work provides a too basic proof-of-concept for implantable optoelectronic device. In this respect, the work fails to demonstrate sufficient level of integration of the photodetector and the stimulation element (no implantable design or device is presented). I am listing below specific comments that could help the authors to improve their work.*

Our response (1) to summary comments: We sincerely appreciate the reviewer for valuable and critical comments (above and followings) that have significantly improved the quality of our manuscript. First, as the reviewer pointed out, the performance of our device may be similar to that of the previously reported phototransistors using the MoS₂/graphene heterostructure. In past years, many researches have dramatically improved the performance of MoS₂-based devices. Previous researches have mainly focused on improving the single cell performances of MoS₂-based devices and identifying their working mechanism, but a MoS₂-based image sensor array in a hemispherically curved format that can capture 2D optical images has not been reported yet. We developed the ultrathin and extremely soft version of the high-density curved image sensor array using the MoS₂-graphene heterostructure, and it was the first and very important attempt to achieve high-quality imaging without optical aberration by combining the single lens optics. In a device point of view, we solved the density and device fracture issues of the curved image sensors by introducing the ultrathin device structure (51 nm), using inherently soft materials (MoS₂ and graphene), applying a strain-isolation device design (isolation of Al₂O₃ and Si₃N₄), and introducing a truncated icosahedron design. We modified the manuscript to highlight the novelty of our result in terms of optoelectronic devices and systems.

Our modification to the manuscript:

(Line 6, page 2: in revised main text)

“In this study, we describe a high-density and hemispherically curved image sensor (CurVIS) array that leverages the atomically thin MoS₂-graphene heterostructure and strain-releasing device designs.”

(Line 22, page 3: in revised main text)

“The softness^{23,27} and ultrathin thickness^{28,29} of MoS₂ are additional factors that enable the fabrication of the soft optoelectronic device. However, an ultrasoft MoS₂-based multicell optoelectronic device that can capture images on the hemispherical surface and its application to soft bioelectronics have not been reported yet”

(Line 4, page 4: in revised main text)

“Here, we present an ultrasoft and high-density curved MoS₂-graphene photodetector array using single-lens optics.”

(Line 12, page 4: in revised main text)

“The high-density MoS₂-graphene CurvIS array successfully recognizes various projected images without infrared (IR) noise. It is the first attempt to achieve high-quality imaging using the ultrathin MoS₂-based optoelectronic device in a hemispherically curved format with the single lens optics.”

(Line 4, page 16: in revised main text)

“In summary, the high-density MoS₂-graphene CurvIS array is developed by using ultrathin soft materials and strain-isolating/-releasing device designs.”

Our response (2) to summary comments: Secondly, as the reviewer pointed out, our *in vivo* experiment for retinal stimulation may not be at a level compatible with clinical settings with commercialized retinal prosthetic devices such as pioneering works of Alpha-IMS from the Zrenner group and subretinal prosthetic devices from the Palanker group. However, we hope to emphasize our opinion that the main contribution and important novelty of this work are in the development of the ultrathin and extremely soft MoS₂-graphene CurvIS array as a next-generation imaging element for retinal implants. In addition, we proved the importance of the soft optoelectronic device in the retinal implants, whose shape and mechanical stiffness are similar to those of human retina, and demonstrated *in vivo* retinal stimulation as a proof-of-concept. The value of this work is in showing the potential of these technological advances. We modified the manuscript to convey these points more clearly.

Our modification to the manuscript:

(Line 12, page 2: in revised main text)

“Then we propose the ultrathin CurvIS array as a promising imaging element in the soft retinal implant. The CurvIS array is applied as a human-eye-inspired soft implantable optoelectronic device that can detect optical signals and apply programmed electrical stimulation to optic nerves with minimum mechanical side effects to the retina.”

(Line 15, page 4: in revised main text)

“Then we propose a human-eye-inspired soft implantable optoelectronic device consisting of the CurvIS array and ultrathin neural-interfacing electrodes (UNE) by mimicking structural features of the human eye.”

(Line 22, page 8: in revised main text)

“The ultrathin CurvIS array whose shape and mechanical softness are similar to those of the human retina has high potential to be used as a soft photodetecting component in the retinal prosthesis. Hence the developed ultrathin CurvIS array is applied to the human-eye-inspired soft implantable optoelectronic device.”

(Line 17, page 9: in revised main text)

“As shown in Fig. 4b, we propose a soft implantable optoelectronic device by mimicking the structural features of the human eye.”

(Line 13, page 16: in revised main text)

“The proposed human-eye-inspired soft optoelectronic device is a step forward to the next generation soft bioelectronics and the soft imaging element of the retinal prosthesis.”

Our response (3) to summary comments: According to the reviewer’s comment on the integrated system, we newly added an integrated form of soft bioelectronics comprised of the CurvIS array, UNE, and electronics on soft flexible printed circuit board (soft FPCB) for the soft retinal prosthesis in the revised manuscript. We introduced the soft FPCB that interconnects the CurvIS array with the UNE and includes microchips for translating the optical signals from the CurvIS array to the corresponding electric pulses. The soft FPCB achieved soft interfaces with surrounding tissues by the encapsulation of ultrasoft silicone rubber and showed high deformability. We added the detailed explanations in Fig. 6, and the related text was also added. More specific and detailed responses and our modifications to the manuscript were included in our response to the following comment #1.

Our response (4) to summary comments: We presented the soft optoelectronic device as a soft and biocompatible imaging component of the retinal implant, and proved its advantages through experiments and theoretical analysis based on mechanics. The soft optoelectronic device was conformally laminated to the artificial eye model with minimal distortion, while lamination of the flexible film and wafer-based electronics induced significant distortion (Fig. 4c). We also showed that the much smaller stress is induced on the artificial eye model by the lamination of the soft optoelectronic device (Fig. 4d). To highlight the significance of the soft and ultrathin form of the implantable optoelectronic device, we newly added the finite element analysis (FEA) results that theoretically explain our experiment on mechanical deformation of the eye model by the implanted devices. According to the FEA result, the soft form of the retinal implant minimizes the mechanical deformation of the retina and thereby prevents the unwanted immune responses and/or damages to the retina. The CurvIS array also can achieve wide field-of-view by covering the large area of the curved retina surface.

Figure 4c | Micro CT image (left) and magnified image (right) showing deformation of (i) the bare eye model, attached by (ii) the soft optoelectronic device, (iii) a flexible film device, and (iv) wafer-based electronics.

Figure 4d | **Induced** stress by three different implanted devices.

Our modification to the manuscript:
 (Line 10, page 2: in revised main text)

“We corroborate the validity of the proposed soft materials and ultrathin device designs through theoretical modeling and finite element analysis.”

(Line 21, page 4: in revised main text)

“Detailed theoretical modeling and finite element analysis (FEA) are also performed to understand the mechanics of the proposed materials and device designs in the retina, which highlights importance of the softness and the curved shape of the optoelectronic device in the retinal implant.”

(Line 5, page 11: in revised main text)

“Finite element analysis of the soft implantable optoelectronics and the eye model

Although the theoretical analysis in Fig. 2 elaborates the mechanical benefits of the truncated icosahedron design, we assume that the hemispherical dome is rigid. However, the retina and sclera structure is actually soft and thin. To numerically compare the mechanical deformation of the eye model by the implantation of three different types of devices (Fig. 4c), we perform FEA to simulate the strains induced in the soft eye model as well as the implanted devices. Figure 4f plots the distribution of principle strain in the eye model without any device (i; Fig. 4f) and with the lamination of the soft optoelectronic device (ii; Fig. 4f), the flexible film (iii; Fig. 4f), and the wafer-based electronics (iv; Fig. 4f). While the soft optoelectronic device causes no visible distortion, the flexible film and the wafer-based electronics induce significant shape change to

the soft eye model, which is not acceptable in practice. Quantitatively, the maximum strain in the eye model induced by the soft optoelectronic device, the flexible film, and the wafer-based electronics are 1.81%, 15.4%, and 9.68%, respectively.

FEA can also be employed to realistically display the effects of the truncated icosahedron design of the soft optoelectronic device (upper row of Fig. 5 and Supplementary Video 1) in comparison with the untruncated circular-shaped device which has the same thickness and materials with our soft optoelectronic device (lower row of Fig. 5 and Supplementary Video 2). Figure 5a shows the strain distribution in the devices. The maximum strain in the soft optoelectronic device is limited to 0.4% after conformal lamination, whereas that in the circular device can be up to 1.89%, which can cause the fracture of the comprising materials. In addition, much fewer wrinkles are observed in the soft optoelectronic device than the untruncated circular film in both experiments and FEA (Fig. 4c and 5b, respectively). The reddish out-of-contact area means the detachment of the device where the gap between the device and the eye model is beyond the thickness of the device (Fig. 5b). This wrinkle-induced delamination can significantly diminish the conformability of the device on the eye model. On the other hand, Figure 5c offers a striking contrast between the maximum strains in the eye model. The strain in the eye model induced by the soft optoelectronic device is only up to 1.81%, whereas the eye model can be deformed to a maximum strain of 13.1% by the circular device. Such distortion is further visible by cross-sectional views (Fig. 5d). In addition, the 15 μm -thick flexible film (iii; Fig. 4c, 4f) and the 525 μm -thick wafer-based electronics (iv; Fig. 4c, 4f) can induce much larger strain to the eye model because of its larger thickness and rigidity. Therefore, we can conclude that the soft optoelectronic device can significantly improve the conformability to the eye model, diminish the strain induced in the device, and reduce the distortion of the soft eye model.”

(Line 8, page 20: in revised main text)

“**Finite Element Analysis of the soft optoelectronic device and the eye model**

The process of attaching the device film to the eye model was simulated by a commercial software (ABAQUS). In experiments, the devices were attached to the eye model by water evaporation. In FEA, the devices were conformed to the eye model by an externally applied pressure, which was converted from the surface tension of the water by the Young-Laplace equation. After the device attached to the eye model, the pressure was unloaded to simulate the drying process. The tangential interaction between the device and the eye model was frictional and the friction coefficient was set to be 0.05, while the normal interaction was no separation after contact. Four-node shell elements were used to model both the device and the eye model. The device was assumed to be a 1.4 μm -thick PI ($E_{\text{PI}} = 2.5 \text{ GPa}$, $\nu_{\text{PI}} = 0.34$) because of the ultrathin thickness of the soft optoelectronic device. The artificial eye model was modelled to be a bilayer structure consistent with the experiment, *i.e.*, 31 μm -thick softer PDMS ($E_{\text{PDMS1}} = 39.3 \text{ kPa}$, $\nu_{\text{PDMS1}} = 0.49$) to be the inner layer and 65 μm -thick stiffer PDMS ($E_{\text{PDMS2}} = 1.18 \text{ MPa}$, $\nu_{\text{PDMS2}} = 0.49$) to be the outer layer.”

(Figure 4f: in the revised manuscript)

Figure 4f | FEA results of the maximum principal strain in eye model (i) without any device, (ii) with the soft optoelectronic device, (iii) with a flexible film device, and (iv) with wafer-based electronics.

(Figure 5: in the revised manuscript)

Figure 5 | Finite element analysis of the soft optoelectronic device and the eye model. (a) Maximum in-plane principle strain distribution in the soft optoelectronic device (top) and the circular device (bottom). (b) The reddish separated part of the soft optoelectronic device (top) and the circular device (bottom). (c) Maximum in-plane principle strain distribution in the eye models attached by the soft optoelectronic device (top) and the circular device (bottom). (d) Deformed shape of the eye model attached by the soft optoelectronic device (top) and the circular device (bottom) obtained by the FEA.

(Supplementary Video 1: in the revised manuscript)

Supplementary Video 1 | Soft eye model attached by the soft optoelectronic device. The video showing the process that the soft optoelectronic device of truncated icosahedron design conforms to the soft eye model. The color means the distance δ between the device and the eye model.

(Supplementary Video 2: in the revised manuscript)

Supplementary Video 2 | Soft eye model attached by the circular film device. The video showing the process that the circular film device conforms to the soft eye model. The color means the distance δ between the device and the eye model.

Our response (5) to summary comments: In addition to FEA results included in our response (4) to summary comments, we added the long-term immunohistology results. In the previous manuscript, we only showed the short-term (1 week) immunohistology results of the retina implanted with the device. To highlight the long-term biocompatibility, we newly added the long-term (9 weeks) immunohistology results of the retina tissues with the implanted device.

Our modification to the manuscript:
(Line 19, page 10: in revised main text)

“Minimal mechanical disturbance by the soft optoelectronic device is also analysed by comparing the histology results of the device-implanted retina (experiment group) and the normal retina (control group). The soft optoelectronic device implanted in the retina both for the short (1 week) and long (9 weeks) period shows good biocompatibility in comparison with the

control group (normal retina; Fig. 4e and Supplementary Fig. 14). The expression of the fibroblast growth factor 2 (FGF2)³⁸ and the glial fibrillary acidic protein (GFAP)³⁸ in the retina implanted with the soft optoelectronic device show similar tendency with those in the normal retina (Supplementary Fig. 14), which indicates the long-term mechanical and material biocompatibility.”

(Line 23, page 21: in revised main text)

“We also utilized 4',6-diamidino-2-phenylindole (DAPI; Vectors Laboratories, USA), FGF2 (1:200, Santa Cruz biotechnology, USA), and GFAP (1:800, Santa Cruz biotechnology, USA) staining by following the standard protocol to prepare fluorescence imaging data of key factors related with retinal biocompatibility.”

(Supplementary Figure 14: in the revised manuscript)

a [Short-term Biocompatibility : 1 week]

Normal retina

Retina implanted with soft optoelectronics

50 μ m

b [Long-term Biocompatibility : 9 weeks]

Normal retina

Retina implanted with soft optoelectronics

50 μ m

Supplementary Figure 14 | Biocompatibility of the soft optoelectronic device. (a,b) The histological staining data (DAPI, FGF2, and GFAP) and differential interference contrast (DIC) microscope image of the normal retina and the retina implanted with the soft optoelectronic device for the short-term (a) and long-term (b) period.

***Comment #1:** The soft implantable optoelectronic device has not sufficient level of integration of the photosensitive element and the stimulating element. Given the level of advancement in the field (see work by Benfenati in Nature Nanotech and Palanker in Nature Photonics), the device should integrate the photodetector pixels with the stimulating microelectrode pixels. In their experiment, the authors use extraocular detection, and amplifier, and a very basic stimulating device. A real advancement in the field would require integration of these functionalities (as already provided in the state of the art).*

Our response to comment #1: We thank the reviewer for important comments on the system integration. As we mentioned in our response (3) to summary comments, we fabricated an integrated form of the soft bioelectronics consisting of the CurvIS array, UNE, and electronics on soft FPCB. The phototransistors in the CurvIS array and the corresponding stimulation electrodes in the UNE were vertically stacked each other. To interconnect the CurvIS array to UNE, the soft and flexible form of the PCB was newly developed and co-integrated with the CurvIS array and UNE in the revised work. The soft FPCB processed the photocurrent produced from the phototransistor and transferred the programmed electric pulses to the stimulation electrode on the same pixel. The integrated device could measure incoming light without crosstalk on an eye model. We added the detailed explanation and fabrication procedures in the revised manuscript.

Our modification to the manuscript:
(Line 18, page 4: in revised main text)

“A soft and flexible image processing unit is also introduced to construct the fully integrated soft implantable electronic system.”

(Line 17, page 12: in revised main text)

“Flexible electronic system integrating the CurvIS array and UNE

One of the important issues in the retinal implant is how to convert the visual information obtained by the image sensor array to the corresponding electrical signals to be conveyed to the retina via the micro-electrode array³⁹. In commercial retinal implants⁴⁰ (e.g., Argus II, Second Sight), the visual information is recognized by a wearable camera module and translated to the electric signals by a video processor to be transmitted to the intraocular micro-electrode array. The electronic devices that supply power and control the system are usually implanted in the extraocular position due to the spatial limitation, but these rigid and bulky devices may cause immune responses and mechanical damages to the surrounding tissues. A photovoltaic type retinal prosthesis without external power sources has been recently reported^{14,41}, but head-mounted glasses are still needed to transfer the IR beam to Si photovoltaic devices. Therefore, the soft CurvIS array and UNE integrated with the flexible implantable electronics⁴² coated with the soft silicone rubber can be a promising candidate of the soft retinal implant due to minimal mechanical mismatch¹⁵ between the tissue and the implanted device.

Figure 6a and 6b show a schematic and a corresponding optical camera image of the integrated soft electronic system. The photocurrent is generated by each phototransistor of the soft CurvIS array in response to external light, and is amplified by a transimpedance amplifier and an inverter. The micro-controller unit (MCU) measures the amplified signal, processes it, and produces programmed electrical pulses. The pulse electrically stimulates the retina via the electrode stacked with the corresponding phototransistor. The ultrathin soft optoelectronic device array is conformally laminated on the eye model (Fig. 6c). As shown in Fig. 6d, each phototransistor of the CurvIS array and a corresponding electrode in the UNE are vertically stacked in the ultrathin and soft platform. To develop a soft form of the fully implantable system, we introduce the flexible printed circuit board with the soft surface coating (soft FPCB; Fig. 6e). The soft FPCB includes all electronics for image processing as depicted in Fig. 6a (also see Supplementary Fig. 15 and Supplementary Table 4), analyzes the photocurrent produced from the phototransistor, and transfers the programmed electrical pulses to the stimulation electrode integrated in the same pixel (Fig. 6d).

Mechanical flexibility and softness of the soft FPCB is confirmed by experimental analyses. The conventional rigid electronics has the modulus in the range of GPa and shows significant mechanical mismatch to the soft human tissues. The surface of the FPCB is coated with thick silicone rubber whose modulus (~ 50 kPa) is similar to that of human tissues (100 - 1500 kPa)¹. This mechanically-matched material property allows soft and conformal interfaces with surrounding tissues. Figure 6f shows the soft FPCB poked by the tip of a pipet tube. Thick coating of the FPCB with silicone rubber provides the cushion-like surface. The silicone rubber coating also effectively protects the electronic chips from external impact and water exposure. Unlike conventional rigid electronics, the soft FPCB also can be easily deformed (Fig. 6g).

The integrated form of the soft optoelectronic system can successfully recognize the illuminated light and generate programmed electrical pulses. The soft integrated system with the FPCB measures the photocurrent generated at each phototransistor, and delivers electrical

stimulation to the eye model using the integrated electrode. When light is illuminated to the pixel ‘a’ and pixel ‘b’ as shown in Fig. 6c, electrical pulses are selectively generated at the electrode of the pixel ‘a’ and pixel ‘b’, respectively (Fig. 6h white region). Figure 6i shows the magnified electrical signals in Fig. 6h. When light is simultaneously illuminated to the pixel ‘a’ and pixel ‘c’, the soft FPCB processes the measured signals from image sensors and successfully generates electrical pulses on both electrodes of pixel ‘a’ and pixel ‘c’ (Fig. 6h blue region).”

(Line 11, page 26: in revised main text)

“39. Zrenner, E. Fighting blindness with microelectronics. *Sci. Transl. Med.* **5**, 210ps16 (2013).
 40. Humayun, M. S. *et al.* Interim results from the international trial of second sight’s visual prosthesis. *Ophthalmology* **119**, 779-788 (2012).
 41. Mathieson, K. *et al.* Photovoltaic retinal prosthesis with high pixel density. *Nat. Photon.* **6**, 391-397 (2012).
 42. Wei, G. *et al.* Fully integrated wearable sensor arrays for multiplexed in situ perspiration analysis. *Nature* **529**, 509-514 (2016).”

(Figure 6: in the revised manuscript)

Figure 6 | Soft flexible printed circuit board that integrates the CurvIS array with UNE. (a) Schematic drawing of the electronics for detecting the external light (bottom) and for applying the stimulation (top). (b) Optical camera image of the CurvIS array and the UNE on the eye model, which are connected by the soft FPCB. (c) Magnified optical camera image of the vertically stacked the CurvIS array and the UNE. (d) Schematic illustration of the phototransistor (bottom) and the stimulation electrode (top) stacked together and connected via the soft FPCB. (e)

Optical camera image of the soft FPCB. (f,g) Optical camera image of the soft FPCB under poking (f) and bending (g). (h,i) Generated electrical pulses at three different pixels by responding the light on/off (h), and magnified electrical pulse (i).

(Supplementary Figure 15: in the revised manuscript)

Supplementary Figure 15 | Soft flexible printed circuit board (soft FPCB). (a-d) Layout of the soft FPCB showing the comprising components (a), top connection map (b), bottom connection map (c), and merged connection map (d).

(Supplementary Table 4: in the revised manuscript)

Item	Model	Item	Model
MCU1	ATMEGA328P	Y1	8 MHz Resonator
J1	MS920SE-FL27E	D1	LY N971-HL-1
U1, U7, U8	LT1462	R2, R9, R14	ELE-R1608F, 5 M Ω
U5	ltc3200es6-5	R3, R8, R10, R13, R15, R18	ELE-R1608F, 1 M Ω
U11	ltc1983es6-5	R4	ELE-R1608F, 10 k Ω
C1, C4, C8, C9	ELE-C2012, 1 μ F	R5	ELE-R1608F, 330 Ω
C5, C7, C10	ELE-C2012, 10 μ F	R6, R11, R16	ELE-R1608F, 0.5 M Ω
C6	ELE-C2012, 0.1 μ F	R7, R12, R17	ELE-R1608F, 10 M Ω

Supplementary Table 4 | Chip information of the electronic circuit. Detailed information of electronic components in the soft FPCB.

Comment #2: *The authors have chosen to use healthy animals for the in vivo validation of the technology. This referee questions this choice because it is difficult to derive conclusions from the electrical stimulation study.*

Our response to comment #2: We appreciate the reviewer’s constructive comment. The retinal degenerated (RD) animal models were not available in our experiment, and hence we used the healthy rats for the *in vivo* validation of our soft optoelectronic device. Extraocular photodetection and corresponding electrical stimulation to the retina were used to prevent the light applied to the photodetector from interfering with the photoreceptor cells in the healthy retina. Furthermore, similar changes in local field potentials and spikes from the animal brain were observed in the experiments with optical stimuli only (Fig. 7b, 7c) and with electrical stimulations only (Fig. 7d, 7e). Thus, we could conclude that electrical stimulation to the retina was successfully applied. We also added selective stimulation experiments using our new integrated system with the soft FPCB on the eye model as described in our response to comment #1. We modified the manuscript to clearly explain that our experiment with the extraocular detection did not have optical interferences on the rat’s eye and only electrical stimulation was applied to the retina.

Figure 7 | Retinal stimulation by the soft optoelectronic device. (b,c) Measurement of elicited spikes (b) and LFP changes (c) in the visual cortex by optical stimulation. (d,e) Measurement of elicited spikes (d) and LFP changes (e) in the visual cortex by electrical stimulation.

Our modification to the manuscript:

(Line 22, page 14: in revised main text)

“The extraocular imager is used to detect the incoming light signals without causing interferences with healthy photoreceptors. The electrode attached on the retina successfully stimulates the optic nerves (Supplementary Fig. 16a, 16b).”

(Line 12, page 22: in revised main text)

“The extraocular light detection was performed to prevent the activation of the healthy retina and the signal interference by external light.”

Thank you very much again for your insightful comments. We feel that these comments have helped to improve the quality of the manuscript significantly.

Reviewer #3:

***Summary Comments:** The paper entitled "Human-eye-inspired soft optoelectronic device using high-density MoS₂-graphene curved image sensor array" shows an very interesting technological development to improve the performance of the retinal implants. The presented development could improve the sensing capabilities of the current retinal implants, making it more suitable for the patient and reducing the external components such as cameras or lenses. However, the authors does not take into account the integration of this development in a functional retinal implant. The authors should consider the following points to understand how this technology can be integrated in a retinal implant.*

Our response to summary comments: We sincerely appreciate the reviewer for valuable and critical comments (above and followings) that significantly improved the quality of our manuscript. As pointed out by the reviewer, all device components could not be completely integrated in our previous manuscript. Therefore we newly added an integrated form of soft bioelectronics comprised of the CurvIS array, UNE, and electronics on soft flexible printed circuit board (soft FPCB) for the soft retinal prosthesis in the revised manuscript. We introduced the soft FPCB that interconnects the CurvIS array with UNE and includes microchips for translating the optical signals to the corresponding electric pulses. The soft FPCB achieved soft interfaces with surrounding tissues by the encapsulation of ultrasoft silicone rubber and showed high deformability. We also demonstrated that the integrated device could measure incoming light signals without crosstalk on an eye model. We added the detailed explanation and fabrication procedures in the revised manuscript.

Our modification to the manuscript:

(Line 18, page 4: in revised main text)

“A soft and flexible image processing unit is also introduced to construct the fully integrated soft implantable electronic system.”

(Line 18, page 12: in revised main text)

“Flexible electronic system integrating the CurvIS array and UNE

One of the important issues in the retinal implant is how to convert the visual information obtained by the image sensor array to the corresponding electrical signals to be conveyed to the retina via the micro-electrode array³⁹. In commercial retinal implants⁴⁰ (e.g., Argus II, Second Sight), the visual information is recognized by a wearable camera module and translated to the electric signals by a video processor to be transmitted to the intraocular micro-electrode array. The electronic devices that supply power and control the system are usually implanted in the extraocular position due to the spatial limitation, but these rigid and bulky devices may cause immune responses and mechanical damages to the surrounding tissues. A photovoltaic type retinal prosthesis without external power sources has been recently reported^{14,41}, but head-mounted glasses are still needed to transfer the IR beam to Si photovoltaic devices. Therefore, the soft CurvIS array and UNE integrated with the flexible implantable electronics⁴² coated with

the soft silicone rubber can be a promising candidate of the soft retinal implant due to minimal mechanical mismatch¹⁵ between the tissue and the implanted device.

Figure 6a and 6b show a schematic and a corresponding optical camera image of the integrated soft electronic system. The photocurrent is generated by each phototransistor of the soft CurvIS array in response to external light, and is amplified by a transimpedance amplifier and an inverter. The micro-controller unit (MCU) measures the amplified signal, processes it, and produces programmed electrical pulses. The pulse electrically stimulates the retina via the electrode stacked with the corresponding phototransistor. The ultrathin soft optoelectronic device array is conformally laminated on the eye model (Fig. 6c). As shown in Fig. 6d, each phototransistor of the CurvIS array and a corresponding electrode in the UNE are vertically stacked in the ultrathin and soft platform. To develop a soft form of the fully implantable system, we introduce the flexible printed circuit board with the soft surface coating (soft FPCB; Fig. 6e). The soft FPCB includes all electronics for image processing as depicted in Fig. 6a (also see Supplementary Fig. 15 and Supplementary Table 4), analyzes the photocurrent produced from the phototransistor, and transfers the programmed electrical pulses to the stimulation electrode integrated in the same pixel (Fig. 6d).

Mechanical flexibility and softness of the soft FPCB is confirmed by experimental analyses. The conventional rigid electronics has the modulus in the range of GPa and shows significant mechanical mismatch to the soft human tissues. The surface of the FPCB is coated with thick silicone rubber whose modulus (~ 50 kPa) is similar to that of human tissues (100 - 1500 kPa)¹. This mechanically-matched material property allows soft and conformal interfaces with surrounding tissues. Figure 6f shows the soft FPCB poked by the tip of a pipet tube. Thick coating of the FPCB with silicone rubber provides the cushion-like surface. The silicone rubber coating also effectively protects the electronic chips from external impact and water exposure. Unlike conventional rigid electronics, the soft FPCB also can be easily deformed (Fig. 6g).

The integrated form of the soft optoelectronic system can successfully recognize the illuminated light and generate programmed electrical pulses. The soft integrated system with the FPCB measures the photocurrent generated at each phototransistor, and delivers electrical stimulation to the eye model using the integrated electrode. When light is illuminated to the pixel 'a' and pixel 'b' as shown in Fig. 6c, electrical pulses are selectively generated at the electrode of the pixel 'a' and pixel 'b', respectively (Fig. 6h white region). Figure 6i shows the magnified electrical signals in Fig. 6h. When light is simultaneously illuminated to the pixel 'a' and pixel 'c', the soft FPCB processes the measured signals from image sensors and successfully generates electrical pulses on both electrodes of pixel 'a' and pixel 'c' (Fig. 6h blue region)."

(Line 11, page 26: in revised main text)

- “39. Zrenner, E. Fighting blindness with microelectronics. *Sci. Transl. Med.* **5**, 210ps16 (2013).
40. Humayun, M. S. *et al.* Interim results from the international trial of second sight’s visual prosthesis. *Ophthalmology* **119**, 779-788 (2012).
41. Mathieson, K. *et al.* Photovoltaic retinal prosthesis with high pixel density. *Nat. Photon.* **6**, 391-397 (2012).
42. Wei, G. *et al.* Fully integrated wearable sensor arrays for multiplexed in situ perspiration analysis. *Nature* **529**, 509-514 (2016).”

(Figure 6: in the revised manuscript)

Figure 6 | Soft flexible printed circuit board that integrates the CurvIS array with UNE. (a) Schematic drawing of the electronics for detecting the external light (bottom) and for applying the stimulation (top). (b) Optical camera image of the CurvIS array and the UNE on the eye model, which are connected by the soft FPCB. (c) Magnified optical camera image of the vertically stacked the CurvIS array and the UNE. (d) Schematic illustration of the phototransistor (bottom) and the stimulation electrode (top) stacked together and connected via the soft FPCB. (e) Optical camera image of the soft FPCB. (f,g) Optical camera image of the soft FPCB under poking (f) and bending (g). (h,i) Generated electrical pulses at three different pixels by responding the light on/off (h), and magnified electrical pulse (i).

(Supplementary Figure 15: in the revised manuscript)

Supplementary Figure 15 | Soft flexible printed circuit board (soft FPCB). (a-d) Layout of the soft FPCB showing the comprising components (a), top connection map (b), bottom connection map (c), and merged connection map (d).

(Supplementary Table 4: in the revised manuscript)

Item	Model	Item	Model
MCU1	ATMEGA328P	Y1	8 MHz Resonator
J1	MS920SE-FL27E	D1	LY N971-HL-1
U1, U7, U8	LT1462	R2, R9, R14	ELE-R1608F, 5 M Ω
U5	ltc3200es6-5	R3, R8, R10, R13, R15, R18	ELE-R1608F, 1 M Ω
U11	ltc1983es6-5	R4	ELE-R1608F, 10 k Ω
C1, C4, C8, C9	ELE-C2012, 1 μ F	R5	ELE-R1608F, 330 Ω
C5, C7, C10	ELE-C2012, 10 μ F	R6, R11, R16	ELE-R1608F, 0.5 M Ω
C6	ELE-C2012, 0.1 μ F	R7, R12, R17	ELE-R1608F, 10 M Ω

Supplementary Table 4 | Chip information of the electronic circuit. Detailed information of electronic components in the soft FPCB.

***Comment #1:** The main limitation of the retinal implants is in how the optical information recollected by an external camera is transferred efficiently to the nervous system. This means, how to increase the number of stimulation sites and how to make this stimulation more effective (in terms of qualitative recognition by the patient)*

Our response (1) to comment #1: We appreciate the reviewer’s constructive comment. As the reviewer pointed out, we tried to change bulky and rigid optoelectronic components and related electronics to be miniaturized, ultrathin, hemispherically curved, and soft enough to be mechanically compatible with retina and surrounding tissues. In the revised manuscript, therefore, we added the integrated form of the soft bioelectronics consisting of the CurvIS array, UNE, and soft FPCB as we explained in our previous response to summary comments. Each phototransistor in the CurvIS array and the corresponding electrode in the UNE were vertically stacked in the ultrathin and soft platform, laminated to the eye model, and integrated via the external soft FPCB. For the further increase of stimulation sites and miniaturization of the system, the active-matrix form of the image sensor array and electrode array can be used. The previously reported retinal prosthesis (*e.g.*, Alpha-IMS) is a good example of this approach. We will be able to implement the similar strategy in the future.

Our modification to the manuscript:
(Line 14, page 13: in revised main text)

“The ultrathin soft optoelectronic device array is conformally laminated on the eye model (Fig. 6c). As shown in Fig. 6d, each phototransistor of the CurvIS array and a corresponding electrode in the UNE are vertically stacked in the ultrathin and soft platform. To develop a soft form of the fully implantable system, we introduce the flexible printed circuit board with the soft surface coating (soft FPCB; Fig. 6e). The soft FPCB includes all electronics for image processing as depicted in Fig. 6a (also see Supplementary Fig. 15 and Supplementary Table 4), analyzes the photocurrent produced from the phototransistor, and transfers the programmed electrical pulses to the stimulation electrode integrated in the same pixel (Fig. 6d).”

(Line 15, page 15: in revised main text)

“We could capture various images using the ultrathin high-density CurvIS array and single lens optics, and demonstrate the prototype of the soft retinal implant consisting of the CurvIS array, UNE, and soft FPCB. To improve the imaging quality of the soft retinal implant, the large amount of optical information obtained by the large number of image sensors should be effectively processed and transferred to the corresponding stimulation electrodes³⁹. However, the increase in the pixel density is challenging because the number of interconnecting wires is proportional to the number of pixels³⁹. Alpha-IMS³⁷, one of the leading retinal prostheses, can be a good approach to solve this issue. Alpha-IMS³⁷ has 1,500 pixels, each of which contains a photodetector, an integrated circuit, and an electrode. The electronic circuit in the individual pixel minimizes the number of external wires by processing and generating the electrical pulses by itself. The similar strategy can be applied to the soft retinal implant by fabricating the self-processable pixel array composed of the ultrathin photodetector and electrode pair in the future.”

(Line 11, page 26: in revised main text)

“39. Zrenner, E. Fighting blindness with microelectronics. *Sci. Transl. Med.* **5**, 210ps16 (2013).”

Our response (2) to comment #1: Another challenge in retinal implants is how to miniaturize and fully implant the external camera into the retina without biological side effects. And for effective integration of the photodetecting unit with the retina, the long-term biocompatibility of the implanted optoelectronic device is critical. The soft form of the retinal implant minimizes mechanical deformation of the retina, prevents the unwanted immune responses, inflammation, and damages to the retina, and can achieve wide field-of-view by covering the large area of the curved retina surface. This would dramatically improve the efficacy of the prosthesis in long-term implantation. We presented the soft optoelectronic device as a soft and mechanically biocompatible imaging component of the retinal implant, and proved its advantages through experiments and theoretical analysis based on mechanics. The soft optoelectronic device was conformally laminated to the artificial eye model with minimal mechanical distortion, while lamination of the flexible film and wafer-based electronics induced significant distortion (Fig. 4c). We also showed that the much smaller stress is induced on the artificial eye model by the lamination of the soft optoelectronic device (Fig. 4d). In the previous manuscript, however, we only showed analytical solution based on mechanics. We newly added FEA results of the strain distribution on the eye model induced by the soft optoelectronic device to highlight the minimal deformation by the soft and ultrathin optoelectronic device.

Figure 4c | Micro CT image (left) and magnified image (right) showing deformation of (i) the bare eye model, attached by (ii) the soft optoelectronic device, (iii) a flexible film device, and (iv) wafer-based electronics.

Figure 4d | **Induced** stress by three different implanted devices.

Our modification to the manuscript:

(Line 10, page 2: in revised main text)

“We corroborate the validity of the proposed soft materials and ultrathin device designs through theoretical modeling and finite element analysis.”

(Line 21, page 4: in revised main text)

“Detailed theoretical modeling and finite element analysis (FEA) are also performed to understand the mechanics of the proposed materials and device designs in the retina, which highlights importance of the softness and the curved shape of the optoelectronic device in the retinal implant.”

(Line 5, page 11: in revised main text)

“Finite element analysis of the soft implantable optoelectronics and the eye model

Although the theoretical analysis in Fig. 2 elaborates the mechanical benefits of the truncated icosahedron design, we assume that the hemispherical dome is rigid. However, the retina and sclera structure is actually soft and thin. To numerically compare the mechanical deformation of the eye model by the implantation of three different types of devices (Fig. 4c), we perform FEA to simulate the strains induced in the soft eye model as well as the implanted devices. Figure 4f plots the distribution of principle strain in the eye model without any device (i; Fig. 4f) and with the lamination of the soft optoelectronic device (ii; Fig. 4f), the flexible film (iii; Fig. 4f), and the wafer-based electronics (iv; Fig. 4f). While the soft optoelectronic device causes no visible

distortion, the flexible film and the wafer-based electronics induce significant shape change to the soft eye model, which is not acceptable in practice. Quantitatively, the maximum strain in the eye model induced by the soft optoelectronic device, the flexible film, and the wafer-based electronics are 1.81%, 15.4%, and 9.68%, respectively.

FEA can also be employed to realistically display the effects of the truncated icosahedron design of the soft optoelectronic device (upper row of Fig. 5 and Supplementary Video 1) in comparison with the untruncated circular-shaped device which has the same thickness and materials with our soft optoelectronic device (lower row of Fig. 5 and Supplementary Video 2). Figure 5a shows the strain distribution in the devices. The maximum strain in the soft optoelectronic device is limited to 0.4% after conformal lamination, whereas that in the circular device can be up to 1.89%, which can cause the fracture of the comprising materials. In addition, much fewer wrinkles are observed in the soft optoelectronic device than the untruncated circular film in both experiments and FEA (Fig. 4c and 5b, respectively). The reddish out-of-contact area means the detachment of the device where the gap between the device and the eye model is beyond the thickness of the device (Fig. 5b). This wrinkle-induced delamination can significantly diminish the conformability of the device on the eye model. On the other hand, Figure 5c offers a striking contrast between the maximum strains in the eye model. The strain in the eye model induced by the soft optoelectronic device is only up to 1.81%, whereas the eye model can be deformed to a maximum strain of 13.1% by the circular device. Such distortion is further visible by cross-sectional views (Fig. 5d). In addition, the 15 μm -thick flexible film (iii; Fig. 4c, 4f) and the 525 μm -thick wafer-based electronics (iv; Fig. 4c, 4f) can induce much larger strain to the eye model because of its larger thickness and rigidity. Therefore, we can conclude that the soft optoelectronic device can significantly improve the conformability to the eye model, diminish the strain induced in the device, and reduce the distortion of the soft eye model.”

(Line 8, page 20: in revised main text)

“Finite Element Analysis of the soft optoelectronic device and the eye model

The process of attaching the device film to the eye model was simulated by a commercial software (ABAQUS). In experiments, the devices were attached to the eye model by water evaporation. In FEA, the devices were conformed to the eye model by an externally applied pressure, which was converted from the surface tension of the water by the Young-Laplace equation. After the device attached to the eye model, the pressure was unloaded to simulate the drying process. The tangential interaction between the device and the eye model was frictional and the friction coefficient was set to be 0.05, while the normal interaction was no separation after contact. Four-node shell elements were used to model both the device and the eye model. The device was assumed to be a 1.4 μm -thick PI ($E_{\text{PI}} = 2.5 \text{ GPa}$, $\nu_{\text{PI}} = 0.34$) because of the ultrathin thickness of the soft optoelectronic device. The artificial eye model was modelled to be a bilayer structure consistent with the experiment, *i.e.*, 31 μm -thick softer PDMS ($E_{\text{PDMS1}} = 39.3 \text{ kPa}$, $\nu_{\text{PDMS1}} = 0.49$) to be the inner layer and 65 μm -thick stiffer PDMS ($E_{\text{PDMS2}} = 1.18 \text{ MPa}$, $\nu_{\text{PDMS2}} = 0.49$) to be the outer layer.”

(Figure 4f: in the revised manuscript)

Figure 4f | FEA results of the maximum principal strain in eye model (i) without any device, (ii) with the soft optoelectronic device, (iii) with a flexible film device, and (iv) with wafer-based electronics.

(Figure 5: in the revised manuscript)

Figure 5 | Finite element analysis of the soft optoelectronic device and the eye model. (a) Maximum in-plane principle strain distribution in the soft optoelectronic device (top) and the circular device (bottom). (b) The reddish separated part of the soft optoelectronic device (top) and the circular device (bottom). (c) Maximum in-plane principle strain distribution in the eye models attached by the soft optoelectronic device (top) and the circular device (bottom). (d) Deformed shape of the eye model attached by the soft optoelectronic device (top) and the circular device (bottom) obtained by the FEA.

(Supplementary Video 1: in the revised manuscript)

Supplementary Video 1 | Soft eye model attached by the soft optoelectronic device. The video showing the process that the soft optoelectronic device of truncated icosahedron design conforms to the soft eye model. The color means the distance δ between the device and the eye model.

(Supplementary Video 2: in the revised manuscript)

Supplementary Video 2 | Soft eye model attached by the circular film device. The video showing the process that the circular film device conforms to the soft eye model. The color means the distance δ between the device and the eye model.

Our response (3) to comment #1: In addition to FEA results included in our response (2) to comment #1, we also included the long-term immunohistology results. In the previous manuscript, we only showed the short-term (1 week) immunohistology results of the retina implanted with the device. To highlight the long-term biocompatibility, we newly added the long-term (9 weeks) immunohistology results of the retina tissues with the implanted device.

Our modification to the manuscript:
(Line 19, page 10: in revised main text)

“Minimal mechanical disturbance by the soft optoelectronic device is also analysed by comparing the histology results of the device-implanted retina (experiment group) and the normal retina (control group). The soft optoelectronic device implanted in the retina both for the short (1 week) and long (9 weeks) period shows good biocompatibility in comparison with the

control group (normal retina; Fig. 4e and Supplementary Fig. 14). The expression of the fibroblast growth factor 2 (FGF2)³⁸ and the glial fibrillary acidic protein (GFAP)³⁸ in the retina implanted with the soft optoelectronic device show similar tendency with those in the normal retina (Supplementary Fig. 14), which indicates the long-term mechanical and material biocompatibility.”

(Line 23, page 21: in revised main text)

“We also utilized 4',6-diamidino-2-phenylindole (DAPI; Vectors Laboratories, USA), FGF2 (1:200, Santa Cruz biotechnology, USA), and GFAP (1:800, Santa Cruz biotechnology, USA) staining by following the standard protocol to prepare fluorescence imaging data of key factors related with retinal biocompatibility.”

(Supplementary Figure 14: in the revised manuscript)

a [Short-term Biocompatibility : 1 week]

Normal retina

Retina implanted with soft optoelectronics

50 μm

b [Long-term Biocompatibility : 9 weeks]

Normal retina

Retina implanted with soft optoelectronics

50 μ m

Supplementary Figure 14 | Biocompatibility of the soft optoelectronic device. (a,b) The histological staining data (DAPI, FGF2, and GFAP) and differential interference contrast (DIC) microscope image of the normal retina and the retina implanted with the soft optoelectronic device for the short-term (a) and long-term (b) period.

Comment #2: The presented development is just matrix of photo-detectors without any stimulation capability. The in vivo experiment for retinal stimulation is performed with just one stimulation electrode. The authors should discuss and show how the large amount of information recollected by the high-density image sensor can be transmitted to the nervous system. At least, the authors should discuss in how to integrate in the sensor a similar range of stimulation electrodes.

Our response to comment #2: We appreciate the reviewer's valuable comment. In the revised manuscript, we added an integrated form of the soft bioelectronics consisting of the CurvIS array, UNE, and soft FPCB as described in our response to summary comments. We also added demonstration of selective stimulation using our system on the eye model. In the current device design, however, further increase of the pixel density is limited because the number of interconnecting wires is proportional to the number of pixels. The previously reported retinal prosthesis (e.g., Alpha-IMS) can be a good approach to solve this issue. Integration of an imaging element, a stimulation electrode, and processing units into a single pixel enables large information processing with the minimized number of external wires. The ultrathin and soft bioelectronics with high-density imaging elements and electrodes can be designed by adapting the similar strategy as a future work. More specific and detailed responses and our modification to the manuscript were included in our response to summary comments and comment #1.

Our modification to the manuscript:
(Line 15, page 15: in revised main text)

“We could capture various images using the ultrathin high-density CurvIS array and single lens optics, and demonstrate the prototype of the soft retinal implant consisting of the CurvIS array, UNE, and soft FPCB. To improve the imaging quality of the soft retinal implant, the large amount of optical information obtained by the large number of image sensors should be effectively processed and transferred to the corresponding stimulation electrodes³⁹. However, the increase in the pixel density is challenging because the number of interconnecting wires is proportional to the number of pixels³⁹. Alpha-IMS³⁷, one of the leading retinal prostheses, can be a good approach to solve this issue. Alpha-IMS³⁷ has 1,500 pixels, each of which contains a photodetector, an integrated circuit, and an electrode. The electronic circuit in the individual pixel minimizes the number of external wires by processing and generating the electrical pulses by itself. The similar strategy can be applied to the soft retinal implant by fabricating the self-processable pixel array composed of the ultrathin photodetector and electrode pair in the future.”

Comment #3: The authors should also detail the electronic required to read the matrix of photo-detectors, in order to discuss about the complexity of integrate it in a retinal implant (i.e. size and number of required connections).

Our response to comment #3: We thank the reviewer for detailed and constructive comments. We added the detailed configuration and explanation of the electronics for reading the matrix of photo-detectors in the revised manuscript. More detailed description on our responses and modifications can be seen in our response to summary comments.

Our modification to the manuscript:
(Line 9, page 13: in revised main text)

“Figure 6a and 6b show a schematic and a corresponding optical camera image of the integrated soft electronic system. The photocurrent is generated by each phototransistor of the soft CurvIS array in response to external light, and is amplified by a transimpedance amplifier and an inverter. The micro-controller unit (MCU) measures the amplified signal, processes it, and produces programmed electrical pulses. The pulse electrically stimulates the retina via the electrode stacked with the corresponding phototransistor. The ultrathin soft optoelectronic device array is conformally laminated on the eye model (Fig. 6c). As shown in Fig. 6d, each phototransistor of the CurvIS array and a corresponding electrode in the UNE are vertically stacked in the ultrathin and soft platform. To develop a soft form of the fully implantable system, we introduce the flexible printed circuit board with the soft surface coating (soft FPCB; Fig. 6e). The soft FPCB includes all electronics for image processing as depicted in Fig. 6a (also see Supplementary Fig. 15 and Supplementary Table 4), analyzes the photocurrent produced from the phototransistor, and transfers the programmed electrical pulses to the stimulation electrode integrated in the same pixel (Fig. 6d).”

(Figure 6: in the revised manuscript)

Figure 6 | Soft flexible printed circuit board that integrates the CurvIS array with UNE. (a) Schematic drawing of the electronics for detecting the external light (bottom) and for applying the stimulation (top). (b) Optical camera image of the CurvIS array and the UNE on the eye model, which are connected by the soft FPCB. (c) Magnified optical camera image of the vertically stacked the CurvIS array and the UNE. (d) Schematic illustration of the phototransistor (bottom) and the stimulation electrode (top) stacked together and connected via the soft FPCB. (e) Optical camera image of the soft FPCB.

Thank you very much again for your insightful comments. We feel that these comments have helped to improve the quality of the manuscript significantly.

Other minor modifications:

#1 (Line 5, page 1: in revised main text)

“Xiaoliang Qin⁵”

#2 (Line 18, page 1: in revised main text)

“⁵*Onfea Computing LLC, 204 Jackson Street, Newton, MA 02459, USA*”

#3 (Line 1, page 2: in revised main text)

“Soft bioelectronic devices provide new opportunities for next-generation implantable devices due in large to **their soft mechanical nature that leads to minimal tissue damages and immune responses**. However, a soft form of the **implantable** optoelectronic device for optical sensing and retinal stimulation has not been developed yet because of the **bulkiness and rigidity** of conventional imaging modules **and their composing materials**.”

#4 (Line 12, page 3: in revised main text)

“These CurvIS arrays have employed distinctive interconnect designs (*e.g.*, pop-up¹⁸ and/or serpentine-shaped¹⁹ structures) to absorb **bending induced** strains in the rigid silicon-based photodetector array.”

#5 (Line 22, page 3: in revised main text)

“The MoS₂-graphene-based CurvIS array shows much lower induced strain than the fracture strain **of composing materials** because of the ultrathin thickness and softness of 2D materials^{23,30}.”

#6 (Line 9, page 6: in revised main text)

“Theoretical analyses of a flat membrane **fully** conforming to a **rigid** hemispherical surface corroborate the validity of the proposed materials and device designs^{33,34}.”

#7 (Line 15, page 6: in revised main text)

“For quantitative analysis, we adapt simplified models that the truncated icosahedron design is approximated as seven separate small circular films (Fig. 2a top; $R_{fs} = 3.5$ mm), while the **design without cutting** is approximated as a large circular film (Fig. 2a bottom; $R_{fl} = 9.3$ mm).”

#8 (Line 21, page 6: in revised main text)

“Comparing with the **film without cutting**, the overall tensile strain level in the **film with the truncated icosahedron design** is significantly lower in all regions (Fig. 2d).”

#9 (Line 1, page 7: in revised main text)

“Without **the truncated icosahedron design** ($R_{fl}/R_d = 0.82$), maximum radial and hoop strains are as large as 2.86% and 2.77%, respectively.”

#10 (Line 5, page 7: in revised main text)

“With **the truncated icosahedron design** ($R_{fs}/R_d = 0.31$), in contrast, the maximum radial (0.41%) and hoop (0.40%) strains are well below 1%, which ensures mechanical integrity of all materials in the device.”

#11 (Line 8, page 7: in revised main text)

“Without **the truncated icosahedron design**, the high compressive hoop strain near the edge of the film (black line in Fig. 2e) can lead to buckle delamination and self-folding in the device film (Supplementary Fig. 5).”

#12 (Line 12, page 7: in revised main text)

“With **the truncated icosahedron design**, however, the negative hoop strain is effectively reduced (red line in Fig. 2e) and buckle delamination is hardly observed (Supplementary Fig. 4).”

#13 (Line 12, page 9: in revised main text)

“Recently, retinal prostheses using intraocular image sensors have been reported as alternatives (Supplementary Fig. 12b), but these still suffer from various issues; absence of a **multi-lens** system for focusing images³⁷ and unwanted immune responses caused by non-conformal integration and/or mechanical mismatch¹⁵ between soft **retina** and rigid devices.”

#14 (Line 18, page 9: in revised main text)

“The ultrathin soft optoelectronic device **consisting of the CurvIS array and UNE** is conformally laminated on the hemispherical retina.”

#15 (Line 23, page 9: in revised main text)

“The mechanical mismatch between the implanted device and retina **may apply** continuous pressures to the eye and **cause** neural degradation particularly in long-term implantation^{1,2,8}.”

#16 (Line 3, page 10: in revised main text)

“An artificial retina and sclera **model** (*i.e.*, a double-layered elastomeric hemispherical shell having similar modulus with human eye) is **prepared** to reveal mechanical deformation of the eye **by the** device implantation (Supplementary Fig. 13). As shown in Fig. 4c, the soft **optoelectronic device conforms** to the artificial eye model (i; original model) with minimal deformation (ii; 1.4 μm -thick soft optoelectronic device), while lamination of a flexible film (iii; 15 μm -thick

flexible film) and wafer-based electronics (iv; 525 μm -thick silicon device) induce significant distortion.”

#17 (Line 8, page 16: in revised main text)

“The CurvIS array and UNE are integrated **through the soft FPCB** to form a human-eye-inspired soft implantable optoelectronic device, which causes minimal mechanical deformation **to** the eye model **as validated by both experiments and corresponding FEA simulations.**”

#18 (Line 18, page 21: in revised main text)

“**Antibiotics** and dexamethasone **were treated** to prevent inflammation by the surgical procedures.”

#19 (Line 8, page 22: in revised main text)

“**The detailed** experimental setup is shown in Supplementary Fig. 16.”

#20 (Line 10, page 22: in revised main text)

“A commercial white LED was fixed in front of the eyeball and delivered **light** for the optical stimulation.”

#21 (Line 1, page 27: in revised main text)

“This research was supported by **IBS-R006-A1** and US NSF CMMI-1541684.”

#22 (Line 6, page 27: in revised main text)

“S.L., **X. Q.**, and N.L. performed theoretical analysis on mechanics.”

#23 (Figure 4d: in the revised manuscript)

Figure 4d | **Induced** stress by three different implanted devices.

REVIEWERS' COMMENTS:

Reviewer #1 (Remarks to the Author):

Overall, I retain my previous opinion that this manuscript is an impressive and thorough study with promising performance in a specialized application where a flexible device is critical. The authors have made sufficient revisions to address my original concerns, adding more details and explanations. They have also made appropriate revisions to address the other reviewers' concerns: regarding the the points of novelty that distinguish their work from other literature, the additional FEA modeling showing the behaviour of their soft devices and their chosen geometry, new data on biocompatibility of the flexible devices, and the new results on the integrated soft bioelectronic device. The latter is particularly important in demonstrating the utility of their technology. I recommend publication of the manuscript in its present form.

Reviewer #2 (Remarks to the Author):

I have reviewed the new version of the manuscript and the detailed answers of the authors to the comments of all referees. I recognize that that authors have done an excellent work in addressing all comments and questions.

From my point of view, one of the two mains two concerns of this referee remains not solved. The authors continue focusing their manuscript on the bioelectronics device for retinal implants. As the authors write in their abstract:

"Soft bioelectronic devices provide new opportunities for next-generation implantable devices due in large to their soft mechanical nature that leads to minimal tissue damages and immune responses. However, a soft form of the implantable optoelectronic device for optical sensing and retinal stimulation has not been developed yet because of the bulkiness and rigidity of conventional imaging modules and their composing materials."

I fully agree with this sentence. The lack of an implantable optoelectronic device exhibiting great performance results from the difficulty in integrating soft electronics that can transfer (and amplify!) the signal from the photodetector to the stimulating electrodes and then to the retina. This challenge is specific to the technology selected for the photodetector. Given the photodetector proposed by the authors, the solution should demonstrate a suitable integrated system.

Unfortunately, the manuscript does not provide an integrated solution. I do not consider that the hybrid PCB described in the new version of the manuscript is such solution. Coating this PCB with PDMS does not make much sense... the mechanical properties of the coated PCB are not governed by the PDMS but by the PCB material.

In short, I cannot support the publication of the manuscript. It continues to have an excessive focus on the bioelectronics device application, but it still fails to demonstrate the real potential of the new technology concept; such demonstration would require a real integration of the matrix of sensors with the matrix of stimulators. The real integration goes well beyond the PCB solution proposed in the new version of the manuscript.

Reviewer #3 (Remarks to the Author):

In the revised version of the manuscript entitled "Human eye-inspired soft optoelectronic device using an array of curved MoS₂-graphene high density image sensors", the authors have addressed the main points to understand the actual applicability of the presented work.

* With respect to the stimulation capabilities of the presented technology, the authors have presented a solution to integrate a stimulation electrode together with each photo-detector. This approach has been validated with a 3-pixel system that is far from the large number of pixels presented in the first part of the work. However, it is valuable information to understand how to

arrive at a realistic implant with the technology presented.

* As for the electronics needed to acquire the signal from the photodetectors, the authors have also added an electronic to control 3 pixels, which control the stimulation electrodes. The electronics presented are too large to be used in an actual implant, however, they can be easily miniaturized by developing an ASIC.

The improvements presented in the manuscript describe how the technology presented can be translated into an actual implant. So I recommend the publication after the minor revision noted below:

* The authors should also add a discussion about the connectivity of the high density matrix with the electronics needed to control it. Taking into account the large number of photodetectors presented, the connectivity between the electronics and the sensor matrix seems to be the main limitation to implement it in an implant.

Reviewer #1:

Summary Comments: Overall, I retain my previous opinion that this manuscript is an impressive and thorough study with promising performance in a specialized application where a flexible device is critical. The authors have made sufficient revisions to address my original concerns, adding more details and explanations. They have also made appropriate revisions to address the other reviewers' concerns: regarding the the points of novelty that distinguish their work from other literature, the additional FEA modeling showing the behaviour of their soft devices and their chosen geometry, new data on biocompatibility of the flexible devices, and the new results on the integrated soft bioelectronic device. The latter is particularly important in demonstrating the utility of their technology. I recommend publication of the manuscript in its present form.

Our response to summary comments: We thank the reviewer for acknowledging the originality and significance of our work.

Thank you very much again for your insightful comments. We feel that these comments have helped to improve the quality of the manuscript significantly.

Reviewer #2:

Summary Comments: *I have reviewed the new version of the manuscript and the detailed answers of the authors to the comments of all referees. I recognize that that authors have done an excellent work in addressing all comments and questions.*

From my point of view, one of the two mains two concerns of this referee remains not solved. The authors continue focusing their manuscript on the bioelectronics device for retinal implants. As the authors write in their abstract:

“Soft bioelectronic devices provide new opportunities for next-generation implantable devices due in large to their soft mechanical nature that leads to minimal tissue damages and immune responses. However, a soft form of the implantable optoelectronic device for optical sensing and retinal stimulation has not been developed yet because of the bulkiness and rigidity of conventional imaging modules and their composing materials.”

I fully agree with this sentence. The lack of an implantable optoelectronic device exhibiting great performance results from the difficulty in integrating soft electronics that can transfer (and amplify!) the signal from the photodetector to the stimulating electrodes and then to the retina. This challenge is specific to the technology selected for the photodetector. Given the photodetector proposed by the authors, the solution should demonstrate a suitable integrated system. Unfortunately, the manuscript does not provide an integrated solution. I do not consider that the hybrid PCB described in the new version of the manuscript is such solution. Coating this PCB with PDMS does not make much sense... the mechanical properties of the coated PCB are not governed by the PDMS but by the PCB material.

In short, I cannot support the publication of the manuscript. It continues to have an excessive focus on the bioelectronics device application, but it still fails to demonstrate the real potential of the new technology concept; such demonstration would require a real integration of the matrix of sensors with the matrix of stimulators. The real integration goes well beyond the PCB solution proposed in the new version of the manuscript.

Our response to summary comments: We sincerely appreciate the reviewer for valuable and critical comments that improve the quality of our manuscript. As the reviewer pointed out, the integration issue is one of the challenging issues in the implantable electronics including retinal prosthetics. In this paper, the main focus is to implement the soft optoelectronic device using the high-density MoS₂-graphene curved image sensor array. Although our prototype of the soft retinal implant, the 3-pixel-based integrated sensor and actuator system, may not enough compared with commercial retinal prosthetic devices (e.g., Argus II or Alpha-IMS), we hope to emphasize the potential of ultrathin 2D material-based photodetector array as an imaging element in soft implantable retinal prosthesis. Through future research, we will try to implement miniaturized integrated system by applying the novel electronic circuit design. As the reviewer #3 suggested, we believe that the introduction of the application-specific integrated circuit (ASIC) can be the key to solve the integration issues. Using the cutting-edge CMOS technology, the entire circuit would be miniaturized into a small single chip, which will address concerns related with the PCB. We modified the manuscript to convey these points.

Our modification to the manuscript:

(Line 21, page 15: in revised main text)

“Innovation in the circuit design (e.g., application specific integrated circuit; ASIC) can effectively miniaturize the size of the electronic components for the retinal implant.”

Thank you very much again for your insightful comments. We feel that these comments have helped to improve the quality of the manuscript significantly.

Reviewer #3:

Summary Comments: *In the revised version of the manuscript entitled "Human eye-inspired soft optoelectronic device using an array of curved MoS₂-graphene high density image sensors", the authors have addressed the main points to understand the actual applicability of the presented work.*

** With respect to the stimulation capabilities of the presented technology, the authors have presented a solution to integrate a stimulation electrode together with each photo-detector. This approach has been validated with a 3-pixel system that is far from the large number of pixels presented in the first part of the work. However, it is valuable information to understand how to arrive at a realistic implant with the technology presented.*

** As for the electronics needed to acquire the signal from the photodetectors, the authors have also added an electronic to control 3 pixels, which control the stimulation electrodes. The electronics presented are too large to be used in an actual implant, however, they can be easily miniaturized by developing an ASIC.*

The improvements presented in the manuscript describe how the technology presented can be translated into an actual implant. So I recommend the publication after the minor revision noted below:

Our response to summary comments: We sincerely appreciate the reviewer for evaluation of our revised work and valuable comments that significantly improve the quality of our manuscript. The main focus of this paper is to implement the soft optoelectronic device using the high-density MoS₂-graphene curved image sensor array. Our prototype of the soft retinal implant, the 3-pixel-based integrated sensor and actuator system, provides the potential of the ultrathin 2D material-based photodetector array as a soft implantable retinal prosthesis although the integrated system still has some miniaturization and integration issues. As the reviewer suggested, introduction of application-specific integrated circuit (ASIC) can be the key to solve the aforementioned issues. We modified the manuscript to describe these points.

Our modification to the manuscript:

(Line 10, page 16: in revised main text)

“Innovation in the circuit design (e.g., application specific integrated circuit; ASIC) can effectively miniaturize the size of the electronic components for the retinal implant.”

Comment #1: *The authors should also add a discussion about the connectivity of the high density matrix with the electronics needed to control it. Taking into account the large number of photodetectors presented, the connectivity between the electronics and the sensor matrix seems to be the main limitation to implement it in an implant.*

Our response to comment #1: We appreciate the reviewer’s constructive comment. As the reviewer pointed out, we added a brief discussion about the connectivity issue in the revised manuscript.

Our modification to the manuscript:
(Line 1, page 16: in revised main text)

“The miniaturization and integration of the retina prosthesis depends on improving the connectivity between the high-density sensor matrix and electronics.”

(Line 5, page 16: in revised main text)

“The electronic circuit in the individual pixel processes and generates the electrical pulses by itself, and hence it minimizes the number of external wires and provides improved the connectivity between high density matrix of sensors and electronics.”

(Line 10, page 16: in revised main text)

“Innovation in the circuit design (*e.g.*, application specific integrated circuit; ASIC) can also effectively miniaturize the size of the electronic components for the retinal implant.”

Thank you very much again for your insightful comments. We feel that these comments have helped to improve the quality of the manuscript significantly.

Other editorial and minor modifications:

#1 (Line 1, page 2: in revised main text)

“Abstract

Soft bioelectronic devices provide new opportunities for next-generation implantable devices due to their soft mechanical nature that leads to minimal tissue damages and immune responses.”

#2 (Line 19, page 2: in revised main text)

“Introduction”

#3 (Line 6, page 3: in revised main text)

“Conventional wafer-based rigid and planar imaging modules, however, are far from this goal because lamination of planar devices can cause the retinal deformation¹⁴, stiff devices can damage the non-regenerative optic nerves¹⁵, and bulky multi-lens optics is required to focus on the flat image sensor (Supplementary Fig. 1a).”

#4 (Line 4, page 5: in revised main text)

“Results”

#5 (Line 8, page 5: in revised main text)

“Constructing a high-density image sensor array on the hemispherical surface (Supplementary Fig. 4) to achieve optical advantages (Supplementary Fig. 1, 2 and Supplementary Note 1) requires development of a novel soft photodetector array.”

#6 (Line 22, page 6: in revised main text)

“Analytical solutions in Supplementary Note 2 yield the radial and hoop strain distributions for the truncated and untruncated films fully conformed to a hemispherical dome ($R_d = 11.34$ mm), which are plotted as contour plots (Fig. 2b and 2c, respectively) and curves (Fig. 2d and 2e, respectively).”

#7 (Line 1, page 8: in revised main text)

“The transfer curve (I_d - V_g) shows a typical light-sensitive field effect transistor behaviour (Fig. 3b). Under illumination (515 nm), the MoS₂ channel generates a photocurrent whose normalized magnitude (I_d/I_{dark}) is proportional to the illuminated light intensity (Fig. 3c).”

#8 (Line 12, page 10: in revised main text)

“Detailed mechanical analyses regarding the interfacial tractions between three different implantable devices and the eye model are described in Supplementary Note 3.”

#9 (Line 20, page 20: in revised main text)

“The process of attaching the device to the eye model was simulated by a commercial software (ABAQUS). In experiments, the **optoelectronic device was attached to the eye model by water evaporation, while the flexible film and the silicon wafer were attached by finger. To simulate the integration process in FEA, the devices were conformed to the eye model by an externally applied pressure, which was converted from the surface tension of the water by the Young-Laplace equation or from the pressure applied by finger.** After the device attached to the eye model, the **externally applied** pressure was unloaded. The tangential interaction between the device and the eye model was frictional and the friction coefficient was set to be 0.05, while the normal interaction was no separation after contact. Four-node shell elements were used to model **the optoelectronic device, the flexible film, and the eye model. Eight-node solid elements were used to model the silicon wafer.** The **optoelectronic** device was assumed to be a 1.4 μm -thick PI ($E_{\text{PI}} = 2.5 \text{ GPa}$, $\nu_{\text{PI}} = 0.34$) because of the ultrathin thickness of the soft optoelectronic device, **while the flexible film and silicon wafer were set to be a 15 μm -thick Al film ($E_{\text{Al}} = 69 \text{ GPa}$, $\nu_{\text{Al}} = 0.34$) and 525 μm -thick Si wafer ($E_{\text{Si}} = 165 \text{ GPa}$, $\nu_{\text{Si}} = 0.22$), respectively.** The artificial eye model was modelled to be a bilayer structure consistent with the experiment, *i.e.*, 31 μm -thick softer PDMS (**neo-Hookean material, $C_{10,\text{PDMS1}} = 6.55 \text{ kPa}$, $D_{1,\text{PDMS1}} = 0.0122 \text{ kPa}^{-1}$**) to be the inner layer and 65 μm -thick stiffer PDMS (**$C_{10,\text{PDMS2}} = 204 \text{ kPa}$, $D_{1,\text{PDMS2}} = 0.0508 \text{ MPa}^{-1}$**) to be the outer layer.”

#10 (Line 22, page 21: in revised main text)

“In this study, we used male Wistar rats whose weights are in the range of 280–300 g **and in the age of 10–12 weeks** (Japan SLC; Hamamatsu, Japan). The animals were **housed** at the temperature of 22–24 °C with a 12/12 hr light/dark cycle.”

#11 (Line 21, page 23: in revised main text)

“Data availability.

The data files that support the findings of this study are available from the corresponding author on reasonable request.”

#12 (Line 9, page 24: in revised main text)

“4. Maya-Vetencourt, J. F. *et al.* A fully organic retinal prosthesis restores vision in a rat model of degenerative blindness. *Nat. Mater.* **16**, 681-689 (2017).”

#13 (Line 1, page 1: in revised Supplementary information)

“Supplementary Notes”

#14 (Line 3, page 1: in revised Supplementary information)

“Supplementary Note 1. Optic simulation for various optical systems”

#15 (Line 1, page 2: in revised Supplementary information)

“Supplementary Note 2. Theoretical analysis of the soft optoelectronic device based on mechanics”

#16 (Line 20, page 2: in revised Supplementary information)

“Therefore, in the following discussion, we neglect the z/R_d terms in **Supplementary Eq. (1)**.”

#17 (Line 2, page 3: in revised Supplementary information)

“Plugging the r 's into **Supplementary Eq. (1)**, we find that both maximum strains have a quadratic relation with R_f/R_d .”

#18 (Line 10, page 3: in revised Supplementary information)

“Supplementary Note 3. Analytical solution of interfacial tractions between implantable devices and the artificial eye model.”

#19 (Line 1, page 27: in revised Supplementary information)

“Supplementary References

1. Yang, S. & Lu, N. Gauge factor and stretchability of silicon-on-polymer strain gauges, *Sensors* **13**, 8577–8594 (2013).
2. Majidi, C. & Fearing, R. S. Adhesion of an elastic plate to a sphere. *Proc. R. Soc. A* **464**, 1309–1317 (2008).
3. Song, Y. M. *et al.* Digital cameras with designs inspired by the arthropod eye. *Nature* **497**, 95–99 (2013).
4. Jones, I. L., Warner, M. & Stevens, J. D. Mathematical modelling of the elastic properties of retina: a determination of Young's modulus. *Eye* **6**, 556–559 (1992).
5. Ko, M. W. L. Effect of corneal, scleral and lamina cribrosa elasticity, and intraocular pressure on optic nerve damages. *JSM Ophthalmol.* **3**, 1024 (2015).”